# Adversarial Attacks on Graph Classification via Bayesian Optimisation

**Xingchen Wan    Henry Kenlay    Binxin Ru**
**Arno Blaas    Michael A. Osborne    Xiaowen Dong**
Machine Learning Research Group, University of Oxford, Oxford, UK
`{xwan,kenlay,robin,arno,mosb,xdong}@robots.ox.ac.uk`

## Abstract

Graph neural networks, a popular class of models effective in a wide range of graph-based learning tasks, have been shown to be vulnerable to adversarial attacks. While the majority of the literature focuses on such vulnerability in node-level classification tasks, little effort has been dedicated to analysing adversarial attacks on graph-level classification, an important problem with numerous real-life applications such as biochemistry and social network analysis. The few existing methods often require unrealistic setups, such as access to internal information of the victim models, or an impractically-large number of queries. We present a novel Bayesian optimisation-based attack method for graph classification models. Our method is *black-box*, *query-efficient* and *parsimonious* with respect to the perturbation applied. We empirically validate the effectiveness and flexibility of the proposed method on a wide range of graph classification tasks involving varying graph properties, constraints and modes of attack. Finally, we analyse common interpretable patterns behind the adversarial samples produced, which may shed further light on the adversarial robustness of graph classification models. An open-source implementation is available at `https://github.com/xingchenwan/grabnel`.

## 1   Introduction

Graphs are a general-purpose data structure consisting of entities represented by nodes and edges which encode pairwise relationships. Graph-based machine learning models has been widely used in a variety of important applications such as semi-supervised learning, link prediction, community detection and graph classification [3, 51, 14]. Despite the growing interest in graph-based machine learning, it has been shown that, like many other machine learning models, graph-based models are vulnerable to adversarial attacks [33, 17]. If we want to deploy such models in environments where the risk and costs associated with a model failure are high e.g. in social networks, it would be crucial to understand and assess the model stability and vulnerability by simulating adversarial attacks.

Adversarial attacks on graphs can be aimed at different learning tasks. This paper focuses on graph-level classification, where given an input graph (potentially with node and edge attributes), we wish to learn a function that predicts a property of interest related to the graph. Graph classification is an important task with many real-life applications, especially in bioinformatics and chemistry [24, 25]. For example, the task may be to accurately classify if a molecule, modelled as a graph whereby nodes represent atoms and edges model bonds, inhibits HIV replication or not. Although there are a few attempts on performing adversarial attacks on graph classification [10, 23], they all operate under unrealistic assumptions such as the need to query the target model a large number of times or access a portion of the test set to train the attacking agent. To address these limitations, we formulate the adversarial attack on graph classification as a black-box optimisation problem and solve it with Bayesian optimisation (BO), a query-efficient state-of-the-art zeroth-order black-box optimiser. Unlike existing work, our method is query-efficient, parsimonious in perturbations and

35th Conference on Neural Information Processing Systems (NeurIPS 2021)

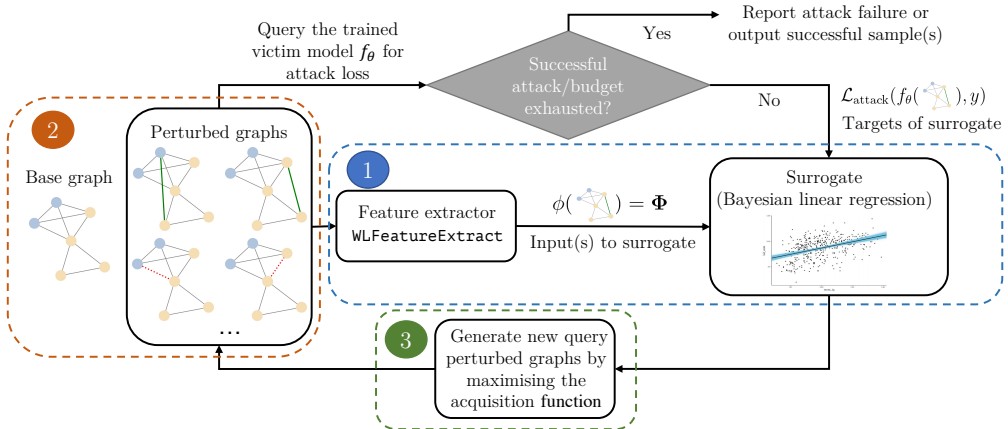

Figure 1: The overall pipeline of GRABNEL. The key components are explained in different paragraphs of Sec 2: *Surrogate model* describes the construction of the BO surrogate and the feature extractor (Block 1), *Sequential perturbation selection* describes how base graphs and perturbed graphs as candidates of adversarial attack are selected (Block 2), and *Optimisation of acquisition function* describes how new query points are generated by BO via optimising acquisition (Block 3). A detailed algorithmic description for GRABNEL is also available in App. A.

does not require policy training on a separate labelled dataset to effectively attack a new sample. Another benefit of our method is that it can be easily adapted to perform various modes of attacks such as deleting or rewiring edges and node injection. Furthermore, we investigate the topological properties of the successful adversarial examples found by our method and offer valuable insights on the connection between the graph topology change and the model robustness.

The main contributions of our paper are as follows. First, we introduce a novel black-box attack for graph classification, GRABNEL[1], which is both query efficient and parsimonious. We believe this is the first work on using BO for adversarial attacks on graph data. Second, we analyse the generated adversarial examples to link the vulnerability of graph-based machine learning models to the topological properties of the perturbed graph, an important step towards interpretable adversarial examples that has been overlooked by the majority of the literature. Finally, we evaluate our method on a range of real-world datasets and scenarios including detecting the spread of fake news on Twitter, which to the best of our knowledge is the first analysis of this kind in the literature.

## 2 Proposed Method: GRABNEL

**Problem Setup**  A graph $\mathcal{G} = (\mathcal{V}, \mathcal{E})$ is defined by a set of nodes $\mathcal{V} = \{v_i\}_{i=1}^n$ and edges $\mathcal{E} = \{\mathbf{e}_i\}_{i=1}^m$ where each edge $\mathbf{e}_k = \{v_i, v_j\}$ connects between nodes $v_i$ and $v_j$. The overall topology can be represented by the adjacency matrix $\mathbf{A} \in \{0, 1\}^{n \times n}$ where $A_{ij} = 1$ if the edge $\{v_i, v_j\}$ is present[2]. The attack objective in our case is to degrade the predictive performance of the trained victim graph classifier $f_\theta$ by finding a graph $\mathcal{G}'$ perturbed from the original test graph $\mathcal{G}$ (ideally with the minimum amount of perturbation) such that $f_\theta$ produces an incorrect class label for $\mathcal{G}$. In this paper, we consider the *black-box evasion* attack setting, where the adversary agent cannot access/modify the the victim model $f_\theta$ (i.e. network architecture, weights $\theta$ or gradients) or its training data $\{(\mathcal{G}_i, y_i)\}_{i=1}^L$; the adversary can only interact with $f_\theta$ by querying it with an input graph $\mathcal{G}'$ and observe the model output $f_\theta(\mathcal{G}')$ as pseudo-probabilities over all classes in a $C$-dimensional standard simplex. Additionally, we assume that *sample efficiency* is highly valued: we aim to find adversarial examples with the minimum number of queries to the victim model. We believe that this is a practical and difficult setup that accounts for the prohibitive monetary, logistic and/or opportunity costs of repeatedly querying a (possibly huge and complicated) real-life victim model. With a high query count, the attacker may also run a higher risk of getting detected. Formally, the objective function of our BO attack agent can be formulated as a black-box maximisation problem:

$$\max_{\mathcal{G}' \in \Psi(\mathcal{G})} \mathcal{L}_{\text{attack}}\big(f_\theta(\mathcal{G}'), y\big) \ \text{ s.t. } \ y = \arg\max f_\theta(\mathcal{G}) \tag{1}$$

---

[1]Stands for *Graph Adversarial attack via BayesiaN Efficient Loss-minimisation*.

[2]We discuss the unweighted graphs for simplicity; our method may also handle other graph types.

where $f_\theta$ is the pretrained victim model that remains fixed in the evasion attack setup and $y$ is the correct label of the original input $\mathcal{G}$. Denote the output logit for the class $y$ as $f_\theta(\mathcal{G})_y$, the *attack* loss $\mathcal{L}_{\text{attack}}$ can be defined as:

$$\mathcal{L}_{\text{attack}}\Big(f_\theta(\mathcal{G}'), y\Big) = \begin{cases} \max_{t \in \mathcal{Y}, t \neq y} \log f_\theta(\mathcal{G}')_t - \log f_\theta(\mathcal{G}')_y & \text{(untargeted attack)} \\ \log f_\theta(\mathcal{G}')_t - \log f_\theta(\mathcal{G}')_y & \text{(targeted attack on class } t\text{)}, \end{cases} \quad (2)$$

where $f_\theta(\cdot)_t$ denotes the logit output for class $t$. Such an attack loss definition is commonly used both in the traditional image attack and the graph attack literature [4, 52] although our method is compatible with any choice of loss function. Furthermore, $\Psi(\mathcal{G})$ refers to the set of possible $\mathcal{G}'$ generated from perturbing $\mathcal{G}$. In this work, we experiment with a diverse modes of attacks to show that our attack method can be generalised to different set-ups:

- creating/removing an edge: we create perturbed graphs by flipping the connection of a small set of node pairs $\delta\mathbf{A} = \{\{u_i, v_i\}\}_{i=1}^{\Delta}$ of $\mathcal{G}$ following previous works [52, 10];
- rewiring or swapping edges: similar to [23], we select a triplet $(u, v, s)$ where we either rewire the edge $(u \to v)$ to $(u \to s)$ (rewire), or exchange the edge weights $w(u, v)$ and $w(u, s)$ (swap);
- node injection: we create new nodes together with their attributes and connections in the graph.

The overall routine of our proposed GRABNEL is presented in Fig 1 (and in pseudo-code form in App A), and we now elaborate each of its key components.

**Surrogate model** The success of BO hinges upon the surrogate model choice. Specifically, such a surrogate model needs to 1) be flexible and expressive enough to locally learn the latent mapping from a perturbed graph $\mathcal{G}'$ to its attack loss $\mathcal{L}_{\text{attack}}(f_\theta(\mathcal{G}'), y)$ (note that this is different and generally easier than learning $\mathcal{G}' \to y$, which is the goal of the classifier $f_\theta$), 2) admit a probabilistic interpretation of uncertainty – this is key for the exploration-exploitation trade-off in BO, yet also 3) be simple enough such that the said mapping can be learned with a small number of queries to $f_\theta$ to preserve sample efficiency. Furthermore, given the combinatorial nature of the graph search space, it also needs to 4) be capable of scaling to large graphs (e.g. in the order of $10^3$ nodes or more) typical of common graph classification tasks with reasonable run-time efficiency. Additionally, given the fact that BO has been predominantly studied in the continuous domain which is significantly different from the present setup, the design of a appropriate surrogate is highly non-trivial. To handle this set of conflicting desiderata, we propose to first use a *Weisfeiler-Lehman (*WL*) feature extractor* to extracts a vector space representation of $\mathcal{G}$, followed by a *sparse Bayesian linear regression* which balances performance with efficiency and gives an probabilistic output.

With reference to Fig. 1, given a perturbation graph $\mathcal{G}'$ as a proposed adversarial sample, the WL feature extractor first extracts a vector representation $\phi(\mathcal{G}')$ in line with the WL subtree kernel procedure (but without the final kernel computation) [30]. For the case where the node features are discrete, let $x^0(v)$ be the initial node feature of node $v \in \mathcal{V}$ (note that the node features can be either scalars or vectors), we iteratively aggregate and hash the features of $v$ with its neighbours, $\{u_i\}_{i=1}^{\deg(v)}$, using the original WL procedure at all nodes to transform them into discrete labels:

$$x^{h+1}(v) = \text{hash}\Big(x^h(v), x^h(u_1), ..., x^h(u_{\deg(v)})\Big), \ \forall h \in \{0, 1, \ldots, H-1\}, \quad (3)$$

where $H$ is the total number of WL iterations, a hyperparameter of the procedure. At each level $h$, we compute the feature vector $\phi_h(\mathcal{G}') = [c(\mathcal{G}', \mathcal{X}_{h1}), ..., c(\mathcal{G}', \mathcal{X}_{h|\mathcal{X}_h|})]^\top$, where $\mathcal{X}_h$ is the set of distinct node features $x^h$ that occur in all input graphs at the current level and $c(G', x^h)$ is the counting function that counts the number of times a particular node feature $x^h$ appears in $G'$. For the case with continuous node features and/or weighted edges, we instead use the modified WL procedure proposed in [36]:

$$x^{h+1}(v) = \frac{1}{2}\Big(x^h(v) + \frac{1}{\deg(v)} \sum_{i=1}^{\deg(v)} w(v, u_i) x^h(u_i)\Big), \ \forall h \in \{0, 1, \ldots, H-1\}, \quad (4)$$

where $w(v, u_i)$ denotes the (non-negative) weight of edge $e_{\{v, u_i\}}$ (1 if the graph is unweighted) and we simply have feature at level $h$ $\phi_h(\mathcal{G}') = \text{vec}(\mathbf{X}_h)$, where $\mathbf{X}_h$ is the feature matrix of graph $\mathcal{G}'$ at level $h$ by collecting the features at each node $\mathbf{X}_h = \Big[x^h(1), ...x^h(v)\Big]$ and $\text{vec}(\cdot)$ denotes the vectorisation operator. In both cases, at the end of $H$ WL iterations we obtain the final feature vector $\phi(\mathcal{G}') = \text{concat}\Big(\phi_1(\mathcal{G}'), ..., \phi_H(\mathcal{G}')\Big)$ for each training graph in $[1, n_{\mathcal{G}'}]$ to form the feature matrix

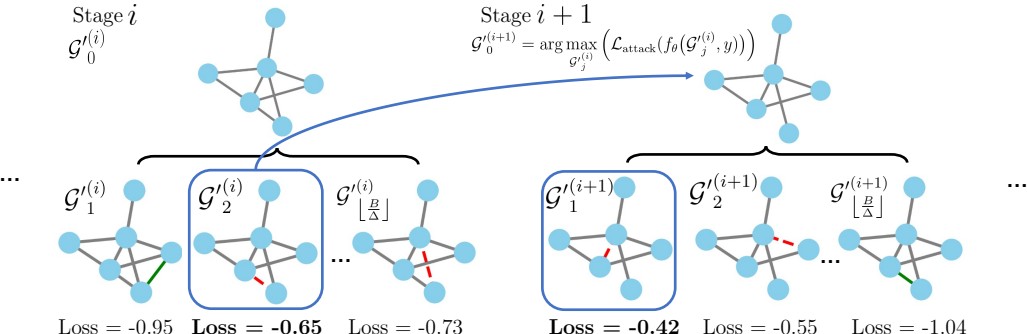

Figure 2: Sequential edge selection. At each stage, the BO agent sequentially proposes candidate graphs with edge edit distance of 1 from the base graph $\mathcal{G}'^{(i)}_0$ (which is the original unperturbed graph $\mathcal{G}$ at initialisation, or a perturbed graph that led to the largest increase in loss from the previous stage otherwise) by selecting the graph that maximises the acquisition function value amongst all candidates generated via sampling/genetic algorithm (see details at *Optimisation of acquisition function*). This procedure repeats until either the attack succeeds (i.e. we find a graph $\mathcal{G}'$ with $\mathcal{L}_{\text{attack}}(f_\theta(\mathcal{G}'), y) > 0$) or the maximum number of $B$ queries to $f_\theta$ is exhausted.

$\mathbf{\Phi} = [\phi(\mathcal{G}'_1), ...\phi(\mathcal{G}'_{|n_{\mathcal{G}'}|})]^\top \in \mathbb{R}^{|n_{\mathcal{G}'}| \times D}$ to be passed to the Bayesian regressor – it is particularly worth noting that the training graphs here denote inputs to train the surrogate model of the attack agent and are typically perturbed versions of a *test* graph $\mathcal{G}$ of the victim model; they are *not* the graphs that are used to train the victim model itself: in an evasion attack setup, the model is considered frozen and the training inputs cannot be accessed by the attack agent any point in the pipeline. The WL iterations capture both information related to individual nodes and topological information (via neighbourhood aggregation), and have been shown to have comparable distinguishing power to some Graph Neural Network (GNN) models [26], and hence the procedure is expressive. Alternative surrogate choices could be, for example, GNNs with the final fully-connected layer replaced by a probabilistic linear regression layer such as the one proposed in [31]. However, in contrast to these, our extraction process $\mathcal{G}' \rightarrow \phi(\mathcal{G}')$ requires no learning from data (we only need to learn the Bayesian linear regression weights) and therefore should lead to better sample efficiency. Alternatively, we may also use a Gaussian Process (GP) surrogate, such as the Gaussian Process with Weisfeiler-Lehman Kernel (GPWL) model proposed in [29] that directly uses a GP model together with a WL kernel. Nonetheless, while GPs are theoretically more expressive (although we empirically show in App. D.1 that in most of the cases their predictive performances are comparable), they are also much more expensive with a cubic scaling w.r.t the number of training inputs. Furthermore, GPWL is designed specifically for neural architecture search, which features small, directed graphs with discrete node features only; on the other hand, the GRABNEL surrogate covers a much wider scope of applications

When we select a large $H$ or if there are many training inputs and/or input graph(s) have a large number of nodes/edges, there will likely be many unique WL features and the resulting feature matrix will be very high-dimensional, which would lead to high-variance regression coefficients $\boldsymbol{\alpha}$ being estimated if $n_{\mathcal{G}'}$ (number of graphs to train the surrogate of the attack agent) is comparatively few. To attain a good predictive performance in such a case, we employ Bayesian regression surrogate with the Automatic Relevance Determination (ARD) prior to learn the mapping $\mathbf{\Phi} \rightarrow \mathcal{L}_{\text{attack}}(f_\theta(\mathcal{G}'), y)$, which regularises weights and encourages sparsity in $\boldsymbol{\alpha}$ [42]:

$$\mathcal{L}_{\text{attack}}|\mathbf{\Phi}, \boldsymbol{\alpha}, \sigma_n^2 \sim \mathcal{N}(\boldsymbol{\alpha}^\top \mathbf{\Phi}, \sigma_n^2 \mathbf{I}), \tag{5}$$

$$\boldsymbol{\alpha}|\boldsymbol{\lambda} \sim \mathcal{N}(\mathbf{0}, \mathbf{\Lambda}), \; \text{diag}(\mathbf{\Lambda}) = \boldsymbol{\lambda}^{-1} = \{\lambda_1^{-1}, ..., \lambda_D^{-1}\}, \tag{6}$$

$$\lambda_i \sim \text{Gamma}(k, \theta) \; \forall i \in [1, D], \tag{7}$$

where $\mathbf{\Lambda}$ is a diagonal covariance matrix. To estimate $\boldsymbol{\alpha}$ and noise variance $\sigma_n^2$, we optimise the model marginal log-likelihood. Overall, the WL routines scales as $\mathcal{O}(Hm)$ and Bayesian linear regression has a linear runtime scaling w.r.t. the number of queries; these ensure the surrogate is scalable to both larger graphs and/or a large number of graphs, both of which are commonly encountered in graph classification (See App D.6 for a detailed empirical runtime analysis).

**Sequential perturbation selection** In the default structural perturbation setting, given an attack budget of $\Delta$ (i.e. we are allowed to flip up to $\Delta$ edges from $\mathcal{G}$), finding exactly the set of perturbations

$\delta\mathbf{A}$ that leads to the largest increase in $\mathcal{L}_{\text{attack}}$ entails an combinatorial optimisation over $\binom{n^2}{\Delta}$ candidates. This is a huge search space that is difficult for the surrogate to learn meaningful patterns in a sample-efficient way even for modestly-sized graphs. To tackle this challenge, we adopt the strategy illustrated in Fig. 2: given the query budget $B$ (i.e. the total number of times we are allowed to query $f_\theta$ for a given $\mathcal{G}$), we assume $B \geq \Delta$ and amortise $B$ into $\Delta$ stages and focus on selecting *one* edge perturbation at each stage. While this strategy is greedy in the sense that it always commits the perturbation leading to the largest increase in loss at each stage, it is worth noting that we do *not* treat the previously modified edges differently, and the agent can, and does occasionally as we observe empirically, "correct" previous modifications by flipping edges back: this is possible as the effect of edge selection is permutation invariant. Another benefit of this strategy is that it can potentially make full use of the entire attack budget $\Delta$ *while* remaining parsimonious w.r.t. the amount of perturbation introduced, as it only progresses to the next stage and modifies the $\mathcal{G}$ further when it fails to find a successful adversarial example in the current stage.

**Optimisation of acquisition function**     At each BO iteration, acquisition function $\alpha(\cdot)$ is optimised to select the next point(s) to query the victim model $f_\theta$. However, commonly used gradient-based optimisers cannot be used on the discrete graph search space; a naïve strategy would be to randomly generate many perturbed graphs, evaluate $\alpha$ on all of them, and choose the maximiser(s) to query $f_\theta$ next. While potentially effective on modestly-sized $\mathcal{G}$ especially with our sequential selection strategy, this strategy nevertheless discards any known information about the search space.

Inspired by recent advances in BO in non-continuous domains [8, 38], we optimise $\alpha$ via an adapted version of the Genetic algorithm (GA) in [10], which is well-suited for our purpose but is not particularly sample efficient since many evolution cycles could be required for convergence. However, the latter is not a serious issue here as we only use GA for acquisition optimisation where we only query the surrogate instead of the victim model, a subroutine of BO that does not require sample efficiency. We outline its ingredients below:

- *Initialisation:* While GA typically starts with random sampling in the search space to fill the initial population, in our case we are not totally ignorant about the search space as we could have already queried and observed $f_\theta$ with a few different perturbed graphs $\mathcal{G}'$. A smoothness assumption on the search space would be that if a $\mathcal{G}'$ with an edge $(u, v)$ flipped from $G$ led to a large $\mathcal{L}_{\text{attack}}$, then another $\mathcal{G}'$ with $(u, s), s \notin \{u, v\}$ flipped is more likely to do so too. To reflect this, we fill the initial population by *mutating* the top-$k$ queried $\mathcal{G}'$s leading to the largest $\mathcal{L}_{\text{attack}}$ seen so far in the current stage, where for $\mathcal{G}'$ with $(u, v)$ flipped from the base graph we 1) randomly choose an end node ($u$ or $v$) and 2) change that node to another node in the graph except $u$ or $v$ such that the perturbed edges in all children shares one common end node with the parent.
- *Evolution*: After the initial population is built, we follow the standard evolution routine by evaluating the acquisition function value for each member as its *fitness*, selecting the top-$k$ performing members as the breeding population and repeating the mutation procedure in initialisation for a fixed number of rounds. At termination, we simply query $f_\theta$ with the graph(s) seen so far (i.e. computing the loss in Fig. 2) with the largest acquisition function value(s) seen during GA.

## 3   Related Works

**Adversarial attack on graph-based models** There has been an increasing attention in the study of adversarial attacks in the context of GNNs [33, 17]. One of the earliest models, Nettack, attacks a Graph Convolution Network (GCN) node classifier by optimising the attack loss of a surrogate model using a greedy algorithm [52]. Using a simple heuristic, DICE attacks node classifiers by adding edges between nodes of different classes and deleting edges connecting nodes of the same class [41]. However, they cannot be straightforwardly transferred graph classification: for Nettack, unlike node classification tasks, we have no access to training input graphs or labels for the victim model during test time to train a similar surrogate in graph classification; for DICE (and also more recent works like [39]), node labels do not exist in graph classification (we only have a single label for the entire graph). We nonetheless acknowledge the other contributions in these works, such as the introduction of constraints to improve imperceptibility, in our experiments in Sec 4.

First methods that do extend to graph classification include [10, 23]: [10] propose a number of techniques, including RL-S2V, which uses reinforcement learning to attack both node and graph classifiers in a black-box manner, and the GA-based attack, which we adapt into our BO acquisition optimisation. However, [10] primarily focus on the S2V victim model, do not emphasise on sample efficiency, and to train a policy that attacks in an one-shot manner on the test graphs, RL-S2V

has to query repeatedly on a separate validation set. We empirically compare against it in App. D.2. Another related work is ReWatt [23], which similarly uses reinforcement learning but through rewiring. Compared to both these methods, GRABNEL does not require an additional validation set and is much more query efficient. Other black-box methods without surrogate models have also been proposed that could be *potentially* be applied to graph classification: [22] exploit common GNN structural bias to attack node features, while [5] relate graph embedding to graph signal processing and construct tailored attack objectives in different GNNs. In comparison to these works that exploit the characteristics of existing architectures to varying degrees, we argue that the optimisation-based method proposed in the our work is more flexible and agnostic to architecture choices, and should be more generalisable to new architectures. Nonetheless, in cases where some architectural information is available, we believe there could be *combinable* benefits: for example, the importance scores proposed in [22] could be used as *sampling weights* as priors to bias GRABNEL towards selecting more vulnerable nodes. We defer detailed investigations of such possibilities to a future work. Finally, there have also been various previous works that focus on a different setup than ours: A white-box optimisation strategy (alternating direction method of multipliers) is proposed in [16]. [48, 44, 2] propose back-door attacks that involve poisoning of the training data before training and/or the test data at inference. [35] attack hierarchical graph pooling networks, but similar to [52] the method requires access to training input/targets. Ultimately, a number of factors, including but not limited to 1) existence/strictness of the query budget, 2) strictness of the perturbation budget, 3) attacker capabilities and 4) sizes of the graphs, would decide which algorithm/setup is more appropriate and should be adopted in a problem-specific way. Nonetheless, we argue that our setup is both challenging and highly significant as it resembles the capabilities a real-life attacker might have (no access to training data; no access to model parameter/gradients and limited query/perturbation budgets).

**Adversarial attacks using BO** BO as a means to find adversarial examples in the black-box evasion setting has been successfully proposed for classification models on tabular [34] and image data [28, 50, 32, 27]. However, we address the problem for graph classification models, which work on structurally and topologically fundamentally different inputs. This implies several nontrivial challenges that require our method to go beyond the vanilla usage of BO: for example, the inputs cannot be readily represented as vectors like for tabular or image data and the perturbations that we consider for such inputs are not defined on a continuous, but on a discrete domain.

## 4 Experiments

We validate the performance of the proposed method in a wide range of graph classification tasks with varying graph properties, including but not limited to the typical TU datasets considered in previous works [10, 23]. As a demonstration of the versatility of the proposed method, instead of considering a single mode of attack which is often impossible in real-life, we also select the attack mode specific to each task. All additional details, including the statistics of the datasets used and implementation details of the victim models and attack methods, are presented in App. C.

**TU Datasets** We first conduct experiments on four common TU datasets [25], namely (in ascending order of average graph sizes in the dataset) IMDB-M, PROTEINS, COLLAB and REDDIT-MULTI-5K. In all cases, unless specified otherwise, we define the attack budget $\Delta$ in terms of the maximum *structural perturbation ratio* $r$ defined in [7] where $\Delta \leq rn^2$. We similarly link the maximum numbers of queries $B$ allowed for individual graphs to their sizes as $B = 40\Delta$, thereby giving larger graphs and thus potentially more difficult instances higher attack[3] and query budgets similar to the conventional image adversarial attack literature [28]. In this work, unless otherwise specified we set $r = 0.03$ for all experiments, and for comparison we consider a number of baselines, including random search, GA introduced in [10][4]. On some task/victim model combinations, we also consider an additional simple gradient-based method which greedily adds or delete edges based on the magnitude computed input gradient similar to the gradient based method described in [10] (note that this method is *white-box* as access to parameter weights and gradients is required), which is also similar in spirit to methods like Nettack [52]. To verify whether the proposed attack method can be used for a variety of classifier architectures we also consider various victim models: we first use GCN [19] and Graph Isomorphism Network (GIN) [45], which are most commonly used in related works [33]. Considering

---

[3]Due to computational constraints, we cap the maximum number of queries to be $2 \times 10^4$ on each graph.

[4]The original implementation of RL-S2V, the primary algorithm in [10], primarily focus on a S2V-based victim model [9]. We compare GRABNEL against it in the same dataset considered in [10] in App. D.2.

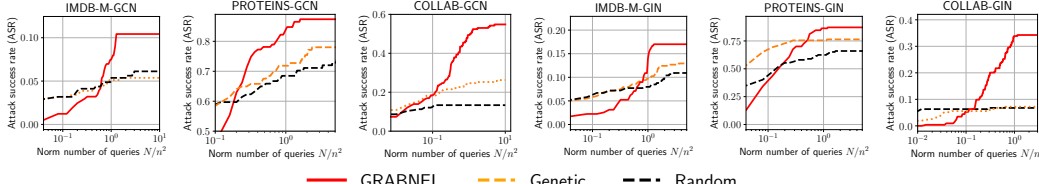

Figure 3: ASR against the number of queries to GCN and GIN (normalised by the square of the number of nodes of each graph) of TU datasets. Note the x-axis is on log-scale. Lines and shades denote mean $\pm 1$ sd across 3 random initialisations. GRABNEL outperforms other attack methods considerably. Random and Genetic appear to converge faster initially as they always use exploit full perturbation budget allocated, while GRABNEL is parsimonious and only attempts a higher perturbation budget when attacks perturbing fewer edges fail. It is also possible to derive a variant of random search with such parsimony, and the readers are referred to the detailed ablation studies in App. D.5.

Table 1: Test accuracy of GCN and GIN victim models on the TU datasets before (*clean*) and after various attack methods. Results shown in mean $\pm$ 1 standard deviation across 3 trials. The results for REDDIT-MULTI-5K are shown in App. D.4.

| | GCN [19] | | | GIN[45] | | |
| | IMDB-M | PROTEINS | COLLAB | IMDB-M | PROTEINS | COLLAB |
|---|---|---|---|---|---|---|
| *Clean* | $50.53_{\pm 1.4}$ | $71.73_{\pm 2.6}$ | $79.73_{\pm 2.1}$ | $48.85_{\pm 0.4}$ | $70.53_{\pm 2.3}$ | $80.80_{\pm 0.9}$ |
| Random | $47.43_{\pm 1.2}$ | $19.46_{\pm 1.7}$ | $67.00_{\pm 3.7}$ | $40.44_{\pm 2.5}$ | $23.21_{\pm 14}$ | $73.01_{\pm 5.0}$ |
| Genetic [10] | $47.82_{\pm 1.5}$ | $14.88_{\pm 1.7}$ | $58.61_{\pm 7.9}$ | $39.68_{\pm 3.1}$ | $15.47_{\pm 10}$ | $72.34_{\pm 2.5}$ |
| Gradient-based† | $\mathbf{39.31}_{\pm \mathbf{2.2}}$ | $50.60_{\pm 4.5}$ | $36.67_{\pm 1.2}$ | $\mathbf{37.56}_{\pm \mathbf{2.2}}$ | $11.90_{\pm 4.4}$ | $\mathbf{54.00}_{\pm \mathbf{2.9}}$ |
| GRABNEL (ours) | $45.23_{\pm 0.2}$ | $\mathbf{10.82}_{\pm \mathbf{2.5}}$ | $\mathbf{35.38}_{\pm \mathbf{9.3}}$ | $38.22_{\pm 3.9}$ | $\mathbf{10.72}_{\pm \mathbf{6.3}}$ | $57.33_{\pm 4.7}$ |

†: White-box method

the strong performance of hierarchical models in graph classification [12, 46], we also conduct some experiments on the Graph-U-Net [12] as a representative of such architectures. We show the classification performance of both victim models before and after attacks using various methods in Table 1, and we show the Attack Success Rate (ASR) against the (normalised) number of queries in Fig. 3. It is worth noting that in consistency with the image attack literature, we launch and consider attacks on the *graphs that were originally classified correctly*, and statistics, such as the ASR, are also computed on that basis. We report additional statistics, such as the evolution of the attack losses as a function of number of queries of selected individual data points in App. D.3.

The results generally show that the attack method is effective against both GCN and GIN models with GRABNEL typically leading to the largest degradation in victim predictions in all tasks, often performing on par or better than *Gradient-based*, a white-box method. It is worth noting that although *Gradient-based* often performs strongly, there is no guarantee that it always does so: first, for general edge flipping problems, *Gradient-based* computes gradients w.r.t. all possible edges (including those that do not currently exist) and an accurate estimation of such high dimensional gradients can be highly difficult. Second, gradients only capture local information and they are not necessarily accurate when used to extrapolate function value beyond that neighbourhood. However, relying on gradients to select edge perturbations constitutes such an extrapolation, as edge addition/deletion is binary and discrete. Lastly, on the tasks with larger graphs (e.g. COLLAB on GCN and GIN), due to the huge search spaces, we find neither random search nor GA could flip predictions effectively except for some "easy" samples already lying close to the decision boundary; GRABNEL nonetheless performs well thanks to the effective constraint of the search space from the sequential selection of edge perturbation, which is typically more significant on the larger graphs.

We report the results on the Graph U-net victim model in Table 2: as expected, Graph U-net performs better in terms of clean classification accuracy compared to the GCN and GIN models considered above, and it also seem more robust to all types adversarial attacks on the PROTEINS dataset. Nonetheless, in terms of relative performance margin, GRABNEL still outperforms both baselines considerably, demonstrating the flexibility and capability for it to conduct effective attacks even on the more complicated and realistic victim models.

Table 2: Test accuracy of Graph U-net [12] on IMDB-M and PROTEINS before and after attack.

| | IMDB-M | PROTEINS |
|---|---|---|
| *Clean* | 55.33 | 79.46 |
| Random | 45.33 | 75.00 |
| Genetic [10] | 44.00 | 75.00 |
| GRABNEL (ours) | **41.33** | **58.80** |

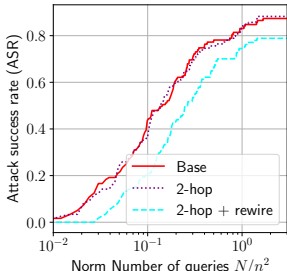
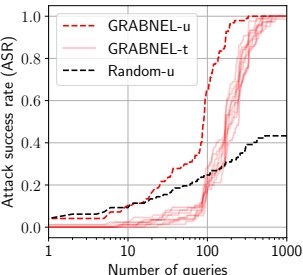
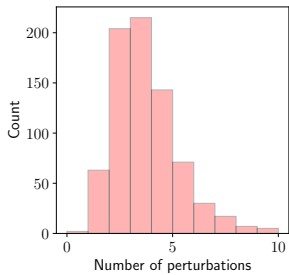

Figure 4: ASR vs normalised # queries with constraints on a GCN model trained on PROTEINS.

Figure 5: ASR vs # queries on a ChebyGIN trained on MNIST-75sp on targeted and untargeted attack setups.

Figure 6: Histogram of #edges swapped in successfully attacked MNIST-75sp instances by GRABNEL-t.

As discussed, in real life, adversarial agents might encounter additional constraints other than the number of queries to the victim model or the amount of perturbation introduced. To demonstrate that our framework can handle such constraints, we further carry out attacks on victim models using identical protocols as above but with a variety of additional constraints considered in several previous works. Specifically, the scenarios considered, in the ascending order of restrictiveness, are:

- Base: The base scenario is identical to the setup in Table 1 and Fig 3;
- 2-hop: Edge addition between nodes $(u, v)$ is only permitted if $v$ is within 2-hop distance of $u$;
- 2-hop+rewire [23]: Instead of flipping edges, the adversarial agent is only allowed to rewire from nodes $(u, v)$ (where an edge exists) to $(u, w)$ (where no edge currently exists). Node $w$ must be within 2-hop distance of $u$;

We test on the PROTEINS dataset, and show the results in Fig. 4: interestingly, the imposition of the 2-hop constraint itself leads to no worsening of performance – in fact, as we elaborate in Sec. 5, we find the phenomenon of adversarial edges remaining relatively clustered within a relatively small neighbourhood is a general pattern in many tasks. This implies that the 2-hop condition, which constrains the spatial relations of the adversarial edges, might already hold even without explicit specification, thereby explaining the marginal difference between the base and the 2-hop constrained cases in Fig. 4. While the additional rewiring constraint leads to (slightly) lower attack success rates, the performance of GRABNEL remains relatively robust in all scenarios considered.

**Image Classification** Beyond the typical "edge flipping" setup on which existing research has been mainly focused, we now consider a different setup involving attacks on the MNIST-75sp dataset [21, 20] with weighted graphs with continuous attributes – note that . The dataset is generated by first partitioning MNIST image into around 75 superpixels with SLIC [1, 11] as the graph nodes (with average superpixel intensity as node attributes). The pairwise distances between the superpixels form the edge weights. We use the pre-trained ChebyGIN with attention model released by the original authors [20] (with an average validation classification accuracy of around 95%) as the victim model. Given that the edge values are no longer binary, simply flipping the edges (equivalent to setting edge weights to 0 and 1) is no longer appropriate. To generalise the sparse perturbation setup and inspired by edge rewiring studied by previous literature, we instead adopt an attack mode via *swapping edges*: each perturbation can be defined by 3 end nodes $(u, v, s)$ where edge weights $w_{uv}$ is swapped with edge weight $w(u, s)$. We show the results in Fig. 5: GRABNEL-u and Random-u denotes the GRABNEL and random search under the *untargeted* attack, respectively, whereas GRABNEL-t denotes GRABNEL under the *targeted attack* with each line denoting 1 of the 9 possible target classes in MNIST. We find that GRABNEL is surprisingly effective in attacking this victim model, almost completely degrading the victim (Fig. 5) with very few swapping operations (Fig. 6) even in the more challenging targeted setup. This seems to suggest that, at least for the data considered, the victim model is very brittle towards carefully crafted edge swapping, with its predictive power seemingly hinged upon a very small number of key edges. We believe a thorough analysis of this phenomenon is of an independent interest, which we defer to a future work.

**Fake news detection** As a final experiment, we consider a real-life task of attacking a GCN-based fake news detector trained on a labelled dataset in [37]. Each discussion cascade (i.e. a chain of tweets, replies and retweets) is represented as an undirected graph, where each node represents a Twitter account (with node features being the key properties of the account such as age and number of followers/followees; see App. C for details) and each edge

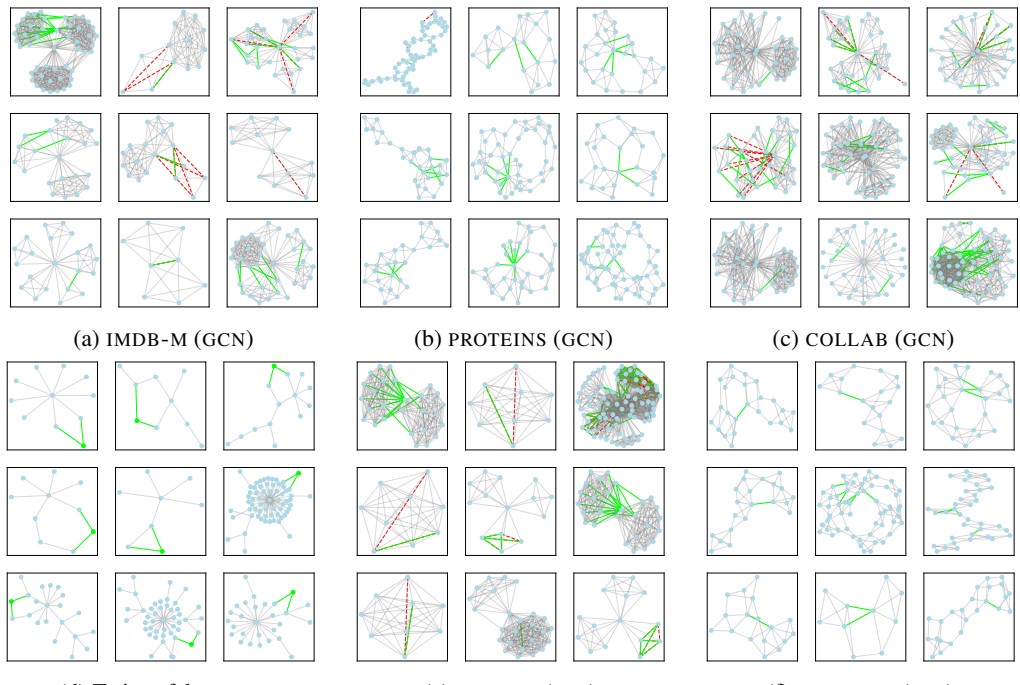

(a) IMDB-M (GCN)   (b) PROTEINS (GCN)   (c) COLLAB (GCN)

(d) Twitter fake news   (e) IMDB-M (GIN)   (f) PROTEINS (GIN)

Figure 7: Adversarial examples found by GRABNEL. Red edges denote deleted edges and green edges indicate added ones. In Twitter fake news detection task, green nodes/edges denote the injected nodes and their connections to the existing graphs. Refer to App D.3 for more examples.

represents a reply/retweet. As a reflection of what a real-life adversary may and may not do, we note that modifying the connections or properties of the existing nodes, which correspond to modifying existing accounts and tweets, is considered impractical and prohibited. Instead, we consider a *node injection* attack mode (i.e. creating new malicious nodes and connect them to existing ones): injecting nodes is equivalent to creating new Twitter accounts and connecting them to the rest of the graph is equivalent to retweeting/replying existing accounts. We limit the maximum number of injected nodes to be $0.05N$ and the maximum number of new edges that may be created per each new node is set to the average number of edges an existing node has – in this context, this limits the number of re-tweets and replies the new accounts may have to avoid easy detection. For the injected node, we initialise its node features in a way that reflects the characteristics of a new Twitter user (we outline the detailed way to do so in App. C). We show the result in Fig. 8, where GRABNEL is capable of reducing the effectiveness of a GCN-based fake news classifier by a third. In this case *Random* also performs reasonably well, as the discussion cascade is typically small, allowing any adversarial examples to be exhaustively found eventually.

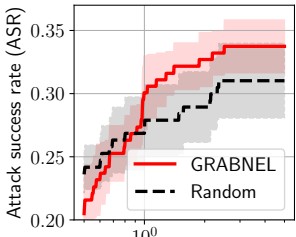

Figure 8: ASR vs. #queries (normalised by the number of nodes, since the attack involves node injection) on the Twitter dataset.

**Ablation Studies**   GRABNEL benefits from a number of design choices and it is important to understand the relative contribution of each to the performance. We find that in *some* tasks GRABNEL without surrogate (i.e. random search with sequential perturbation selection. We term this variant *SequentialRandom*) is a very strong baseline in terms of final ASR, although the full GRABNEL is much better in terms of overall performance, sample efficiency and the ability to produce successful examples with few perturbations. The readers are referred to our ablation studies in App. D.5.

**Runtime Analysis**   Given the setup we consider (sample-efficient black-box attack with minimal amount of perturbation), the cost of the algorithm should not only be considered from the viewpoint of the computational runtime of the attack algorithm itself alone, and this is a primary reason why we use the (normalised) number of queries as the main cost criterion. Nonetheless, a runtime analysis is still informative which we provide in App. D.6. We find that GRABNEL maintains a reasonable

overhead even on, e.g., graphs with $\sim 10^3$ nodes/edges that are larger than most graphs in typical graph classification tasks.

## 5 Attack Analysis

Having established the effectiveness of our method, in this section we provide a qualitative analysis on the common interpretable patterns behind the adversarial samples found, which provides further insights into the robustness of graph classification models against structural attacks. We believe such analysis is especially valuable, as it may facilitate the development of even more effective attack methods, and may provide insights that could be useful for identification of real-life vulnerabilities for more effective defence. We show examples of the adversarial samples in Fig. 7 (and Fig. 13 in App. D.3). We summarise some key findings below.

- *Adversarial edges tend to cluster closely together*: We find the distribution of the adversarial edges (either removal or addition) in a graph to be highly uneven, with many adversarial edges often sharing common end-nodes or having small spatial distance to each other. This is empirically consistent with recent theoretical findings on the stability of spectral graph filters in [18]. From an attacker point of view, this may provide a "prior" on the attack to constrain the search space, as the regions around existing perturbations should be exploited more; we leave a practical investigation of the possibility of leveraging this to enhance attack performance to a future work.

- *Adversarial edges often attempt to destroy or modify community structures*: for example, the original graphs in the IMDB-M dataset can be seen to have community structure, a graph-level topological property that is distinct from the existing works analysing attack patterns on node-level tasks [43, 53]. When the GCN model is attacked, the attack tends to flip the edges *between* the communities, and thereby destroying the structure by either merging communities or deleting edges within a cluster. On the other hand, the GIN examples tend to strengthen the community structures by adding edges within clusters and deleting edges between them. With similar observations also present in, for example, PROTEINS dataset, this may suggest that the models may be fragile to modification of the community structure.

- *Beware the low-degree nodes!* While low-degree nodes are important in terms of degree centrality, we find some victim models are vulnerable to manipulations on such nodes. Most prominently, in the Twitter fake news example, the malicious nodes almost never connect directly to the central node (original tweet) but instead to a peripheral node. This finding corroborates the theoretical argument in [18] which shows that spectral graph filters are more robust towards edge flipping involving *high-degree* nodes than otherwise, and is also consistent with observations on node-level tasks [53] with the explanation being lower-degree nodes having larger influence in the neighbourhood aggregation in GCN. Nonetheless, we note that changes in a higher-degree node are likely to cascade to more nodes in the graph than low degree nodes, and since graph classifiers aggregate across all nodes in the readout layer the indirect change of node representations also matter. Therefore, we argue that this phenomenon in graph classification is still non-trivial.

## 6 Conclusion

**Summary**   This work proposes a novel and flexible black-box method to attack graph classifiers using Bayesian optimisation. We demonstrate the effectiveness and query efficiency of the method empirically. Unlike many existing works, we qualitatively analyse the adversarial examples generated. We believe such analysis is important to the understanding of adversarial robustness of graph-based learning models. Finally, we would like to point out that a potential negative social impact of our work is that bad actors might use our method to attack real-world systems such as a fake new detection system on social media platforms. Nevertheless, we believe that the experiment in our paper only serves as a proof-of-concept and the benefit of raising awareness of vulnerabilities of graph classification systems largely outweighs the risk.

**Limitations and Future Work**   Firstly, the current work only considers topological attack, although the surrogate used is also compatible with attack on node/edge features or hybrid attacks. Secondly, while we have evaluated several mainstream victim models, it would also be interesting to explore defences against adversarial attacks and to test GRABNEL in robust GNN setups such as those with advanced graph augmentations [47], randomised smoothing [48, 13] and adversarial detection [6]. Lastly, the current work is specific to graph classification; we believe it is possible to adapt it to attack other graph tasks by suitably modifying the loss function. We leave these for future works.

## Acknowledgement and Funding Disclosure

The authors would like to acknowledge the following sources of funding in direct support of this work: XW and BR are supported by the Clarendon Scholarship at University of Oxford; HK is supported by the EPSRC Centre for Doctoral Training in Autonomous Intelligent Machines and Systems EP/L015897/1; AB thanks the Konrad-Adenauer-Stiftung and the Oxford-Man Institute of Quantitative Finance for their support. The authors would also like to thank the Oxford-Man Institute of Quantitative Finance for providing the computing resources necessary for this project. The authors declare no conflict in interests.

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
