# APPENDICES of *Adversarial Attacks on Graph Classification via Bayesian Optimisation*

## A  Algorithms

The overall algorithm of GRABNEL is shown in Algorithm 1.

---

**Algorithm 1** Overall pseudocode of the GRABNEL routine.

---

1: **Input:** Original graph $\mathcal{G}_0$, victim model $f_\theta$, $n_{\text{init}}$ (the number of random initialising points), Query budget $B$, Perturbation budget $\Delta$.
2: **Output:** An adversarial graph $\mathcal{G}^*$
3: Set base graph $\mathcal{G}_{\text{base}} \leftarrow \mathcal{G}_0$; initialise stage count `stage` $\leftarrow 0$.
4: Randomly sample $n_{\text{init}}$ perturbed graphs $\{\mathcal{G}'\}_{i=1}^{n_{\text{init}}}$ that are 1 edit distance different from $\mathcal{G}$ and query each perturbed graph to obtain their attack losses $\mathcal{L}_{\text{attack}}(f_\theta, \mathcal{G}'_i)$ (these random samples are counted towards the query budget of the current stage).
5: Compute the WL feature encoding for all graphs: $(\Phi(\mathcal{G}'_1), \ldots, \Phi(\mathcal{G}'_{n_{\text{init}}})) = \texttt{WLFeatureExtract}(\mathcal{G}_0, (\mathcal{G}'_1, \ldots \mathcal{G}'_{n_{\text{init}}}))$.    *// See App. B for details of WLFeatureExtract.*
6: Fit the sparse Bayesian linear regression surrogate with the data $\{\Phi(\mathcal{G}'_i), \mathcal{L}_{\text{attack}}(f_\theta, \mathcal{G}'_i)\}_{i=1}^{n_{\text{init}}}$
7: Divide total budget of $B$ into $\Delta$ stages    *// See "Sequential perturbation selection"*
8: **while** query budget is not exhausted and attack has not succeeded **do**
9:   **if** query budget of the *current stage* is exhausted **then**
10:    Increment the stage count `stage` $\leftarrow$ `stage` $+ 1$ and update the base graph $\mathcal{G}_{\text{base}}$ with the graph leading to largest increase in attack loss in the previous stage.    *// Refer to Fig. 1*
11:   **end if**
12:   Propose graph to be queried next $\mathcal{G}'_{\text{proposal}}$ via acquisition optimisation.    *// See "Optimisation of acquisition function"*
13:   Query $f_\theta$ for the graph proposed in the previous step to calculate its attack loss.
14:   **if** attack succeeded **then**
15:     Set $\mathcal{G}^* \leftarrow \mathcal{G}'_{\text{proposal}}$ and **return** it.
16:   **end if**
17:   Augment the observed data: $\mathcal{D} \leftarrow \mathcal{D} \cup \{\mathcal{G}'_{\text{proposal}}, \mathcal{L}_{\text{attack}}(f_\theta, \mathcal{G}'_{\text{proposal}})\}$, update the WL feature encodings of *all* observed graphs $(\Phi(\mathcal{G}'_1), \ldots, \Phi(\mathcal{G}'_{|\mathcal{D}|})) = \texttt{WLFeatureExtract}(\mathcal{G}_0, (\mathcal{G}'_1, \ldots \mathcal{G}'_{|\mathcal{D}|}))$ and re-fit the surrogate.
18: **end while**
19: **return** None    *// Failed attack within the query budget*

---

## B  WL feature extractor

In this section we describe `WLFeatureExtract` in Algorithm 1 in greater detail. The module takes in both the input graph itself and the set of all input graphs (including itself), as the second argument is to construct a collection of all WL features seen in any of the input graphs and controls the dimensionality of the output feature vector so that the entries in feature vectors of different input graphs represent the same WL feature. For an illustrated example of the procedure, the readers are referred to Fig. 9.

## C  Implementation Details

**Datasets**   We provide some key descriptive statistics of the TU datasets [25] in Table 4. All TU datasets may be downloaded at `https://chrsmrrs.github.io/datasets/docs/datasets/`. The MNIST-75sp dataset is generated from scripts available at `https://github.com/bknyaz/graph_attention_pool`, and the ER-graphs dataset used in App. D.2 is available at `https://github.com/Hanjun-Dai/graph_adversarial_attack`. The details on the Twitter fake news data are described in the following section.

**Twitter dataset**   We used the Twitter dataset described in [37]. In this dataset, each graph represents a rumour cascade. A cascade is made up of nodes which represents both a tweet and the corresponding user who posted the tweet. An edge exists between nodes $u$ and $v$ if $u$ is a retweet of node $v$. The graphs are directed but for simplicity we drop the direction of edges. We use node features which are described in table 3. We apply a log transform to features with a high level of skewness. All data was normalised by subtracting the mean and divided by the standard deviation (estimated using node features from the training set). Further information of these features are give in the supplementary material of [37]. The graph is labelled as true, false or mixed, corresponding to the judged veracity of the rumour, and the learning task is to correctly predict the label of the graph. Many of the graphs in the original dataset are small (and therefore many were topologically similar), we discard samples where the graph has less than $5$ nodes. Furthermore, we use downsampling to balance the dataset so each label appears an equal number of times. After all preprocessing steps are complete the dataset is made up of $4746$ labelled graphs.

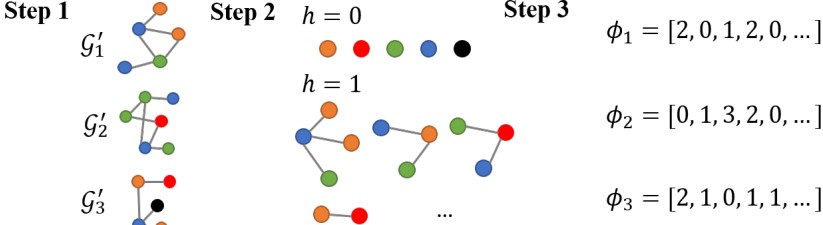

Figure 9: Illustration of the WL extractor. Consider an example of an input of three graphs to the extractor $\{\mathcal{G}_1', \mathcal{G}_2', \mathcal{G}_3'\}$ with colors representing the different (discrete) node labels and we would like to compute `WLFeatureExtract`$(\{\mathcal{G}_i', \{\mathcal{G}_1', \mathcal{G}_2', \mathcal{G}_3'\}) \forall i$ (**Step 1**). The extractor module takes in 2 arguments, as the second argument consisting of the set of all input graphs is used to generate a collection of all possible Weisfeiler-Lehman features seen in *all input graphs* (**Step 2**), up to $H \in \mathbb{N}^+$ where $H$ is the number of WL iterations specified. This step involves computing the Weisfeiler-Lehman embedding on all of the input graphs using the routine introduced in [30]. The extractor finally counts the number of each features present from Step 2 and outputs the feature vector (**Step 3**; only $h = 0$ part of the feature vector is shown in the figure – note that $\mathcal{G}_1'$ has 2 orange nodes, 2 blue nodes and 1 green nodes which yields the corresponding feature vector $\phi_1$). Note that if a particular feature present in the entire set of input graphs is not present in a particular graph, the entry is filled with zero. *The graphs here are for illustration only; in our task each input graph is only one edit distance different from the base graph $\mathcal{G}_0$.*

Table 3: Node features used in the Twitter dataset.

| Feature | Datatype | Transform applied | Description |
|---|---|---|---|
| Rumour category | categorical | One hot encoding | Topic of the rumour |
| Tweet date | float | | Date and time the tweet was posted |
| User account age | float | | How old the account is in days |
| User verified | bool | | If the user is verified by Twitter |
| User followers | int | Log transform | How many followers the user has |
| User followees | int | Log transform | How many other users the user follows |
| User engagement | float | Log transform | How active the user has been since joining |
| Was retweeted | bool | | If the tweet was retweeted |

A challenge of node injection is deciding how to choose node features. We reasoned about how to do this for each feature based on the feature semantics. We choose values based on the assumption that the inserted node (tweet) is being inserted from a bot account trying to emulate other users in the cascade. The rumour category is the same for every node in the graph, so inserted nodes used the same as other nodes in the graph. We set the tweet date to the largest of other nodes, to represent an attack in the evasion setting. We set the user account age, followers, followees, retweet status and verified status to the minimum of all other nodes in the graph (using the convention that *False* ≤ *True*). The user engagement was set to the median among other nodes in the graph. In practice, a user may have control over some of these variables such as the number of followees (by following other accounts) or user engagement (by posting tweets).

**Computing Environment** We conduct all experiments, unless otherwise specified, on a shared server with an Intel Xeon CPU and 256GB of RAM.

**Victim models** We focus our attack on two widely used graph neural networks, namely graph convolutional network (GCN) [19] and graph isomorphism network (GIN) [45]. We also consider an attack on ChebyGIN [20] and Graph U-Net [12]. The graph convolution layers in these models work by aggregating information across the graph edges and then updating combined node features to output new node features. Multiple layers of graph convolution are used. A readout layer transforms the final node embeddings into a fixed-sized graph embedding which can then be fed through a linear layer and a softmax activation function to provide predicted probabilities for each class.

Table 4: Key statistics of the TU datasets used.

| Dataset | #graphs | #labels | Avg #nodes | Avg #edges |
|---------|---------|---------|------------|------------|
| IMDB-M | 1500 | 3 | 13.0 | 65.9 |
| PROTEINS | 1113 | 2 | 39.1 | 72.8 |
| COLLAB | 5000 | 3 | 74.5 | 2457.8 |
| REDDIT-MULTI-5K | 4999 | 5 | 508.8 | 594.9 |

The GCN graph convolutions take the form

$$\mathbf{X}^{(h)} = \sigma(\tilde{\mathbf{D}}^{-1/2}\tilde{\mathbf{A}}\tilde{\mathbf{D}}^{-1/2}\mathbf{X}^{(h-1)}\mathbf{\Theta}^{(h)})$$

where $\tilde{\mathbf{A}} = \mathbf{A} + \mathbf{I}_n$ is the adjacency matrix with self loops, $\tilde{\mathbf{D}} = \text{diag}(d_1 + 1, d_2 + 1, \ldots d_n + 1) = \text{diag}(\mathbf{1}\tilde{\mathbf{A}})$, is a diagonal matrix where $d_u$ is the degree of node $u$. $\mathbf{\Theta}^{(h)}$ and $\mathbf{X}^{(h)}$ are the weight matrix and node features in layer $h$, respectively. For the first layer $\mathbf{X}^{(0)}$ is the original node features. We use three GCN convolutions where the dimension of the hidden node representations are 16. A max pooling across feature maps is applied to the final layer to give a fixed length graph representation which is then used as input to a linear layer.

The graph isomorphism architecture (GIN) is provably more expressive (in terms of distinguishing graph topologies) than the GCN architecture [45]. The graph convolution takes the form

$$\mathbf{X}^{(h)} = \text{MLP}^{(h)}\big((1 + \epsilon^{(h)})\mathbf{X}^{(h-1)} + \mathbf{A}\mathbf{X}^{(h-1)}\big),$$

where $\epsilon^{(h)}$ is a learnable scalar parameter and $\text{MLP}^{(h)}$ is a multilayer perceptron. In our experiments the MLP consists of a single hidden layer of dimension 64 using ReLU activation functions and batch norm applied before applying the activation to the hidden units. We use 5 convolutional layers, applying batchnorm and ReLU activation functions in-between. For the readout function we utilise the representation after each of the GIN convolutions. For each representation, a sum pooling is applied followed by a linear layer. During training dropout is applied to the output of each of the linear layers with probability $p = 0.5$. The outputs of the linear layer are summed to give a final logit score for each class.

The ChebyGIN architecture is similar to the GIN architecture but aggregates information from nodes across multi-hop neighbourhoods. This is achieved by using higher-order Chebyshev polynomials as the aggregation matrix. The polynomial filter is evaluated using the (shifted) normalised Laplacian matrix $\mathbf{L} = -\mathbf{D}^{-1/2}\mathbf{A}\mathbf{D}^{-1/2}$. Chebyshev polynomials can be defined recursively:

$$T_k(\mathbf{L}) = \begin{cases} \mathbf{I}_n, & \text{for } k = 0 \\ \mathbf{L}, & \text{for } k = 1 \\ 2\mathbf{L}T_{k-1}(\mathbf{L}) - T_{k-2}(\mathbf{L}), & \text{for } k > 2 \end{cases} \tag{8}$$

The ChebyGIN convolution is then defined to be

$$\mathbf{X}^{(h)} = \text{MLP}\big((1 + \epsilon)\mathbf{X}^{(h-1)} + T_k(\mathbf{L})\mathbf{X}^{(h-1)}\big).$$

We used a pretrained model used in [20] available at `https://github.com/bknyaz/graph_attention_pool`. We use the model with supervised attention, the best-performing pre-trained model available.

Graph U-Nets are an autoencoder like architecture with skip connections [12]. The architecture consists of a differential graph pooling layer *gPool* and a differential graph unpooling layer *gUnpool* which we briefly describe. To do differential graph pooling a $d \times p$ projection matrix $\mathbf{p}$ is used to compute a length $n$ vector $\mathbf{y} = \mathbf{X}\mathbf{p}$. A ranking operation is used to select the indices of the largest $k$ entries $idx = \text{rank}(\mathbf{y}, k)$ which represent nodes to be included in the sub-graph after pooling. Using this we can select the subgraph adjacency $\mathbf{A}^{(l+1)} = \mathbf{A}^{(l)}[idx, idx]$. The corresponding rows of the features matrix are selected $\tilde{\mathbf{X}}^{(l)} = \mathbf{X}^{(l)}[idx, :]$ and then a re-normalisation is applied to give features for the next layer $\mathbf{X}^{(l+1)} = \tilde{\mathbf{X}}^{(l)} \odot (\text{sigmoid}(\mathbf{y})\mathbf{1}_d^T)$. The *gUnpool* operator does a reverse operation by using the indices computed in the pooling layer and using zero vectors for indices that were not selected during pooling. The architecture uses rounds of pooling and unpooling, as well as skip connections between representations of the same size. For a detailed description of the architecture we refer the reader to [12, Section 3]. We used the open source implementation used in [12]p and provided by the authors at `https://github.com/HongyangGao/Graph-U-Nets`.

**GRABNEL** GRABNEL, which uses the WL feature extractor, involves a number of hyperparameters: the WL procedure is parameterised by a single hyperparameter $H$, which specifies the number of Weisfiler-Lehman iterations to perform. While it is possible for $H$ to be selected automatically via, for example, maximising the log-marginal likelihood of the surrogate model (e.g. [29]), in our case we find fixing $H = 1$ to be performing well. For the sparse Bayesian linear regression model used, we always normalise the input data into hypercubes $[0, 1]^d$ and standardise the target by deducting its mean and dividing by standard deviation. We optimise the marginal log-likelihood via a simple gradient optimiser and we set the maximum number of iterations to be 300. As described in Sec. 2, we need to specify a Gamma prior over over $\{\lambda_i\}$ and we use shape parameter and inverse scale parameters of $1 \times 10^{-6}$. For the acquisition optimisation, we set the maximum number of evaluation of the acquisition function to be 500: we initialise with 50 randomly sampled perturbed graphs, each of which is generated from flipping one pair of randomly selected end nodes from the base graph. To generate the initial population, we fill generate 50 candidates by mutating from the top-3 queried graphs that previously led to the largest attack loss (if we have not yet queried any graphs, we simply sample 100 randomly perturbed graphs). We then evolve the population 10 times, with each evolution cycle involving mutating the current population to generate offspring and popping the oldest members in the population. Finally, we select the top 5 unique candidates seen during the evolution process that have the highest acquisition function value (we use the Expected Improvement (EI) acquisition function) to query the victim model.

## D  Additional Experiments

### D.1  Comparison with Alternative Surrogate Models

We compare the surrogate performance between a GP model with RBF kernel and a Bayesian linear regression model (which is equivalent to GP with linear kernel) to justify the usage of the latter in this section. To make a fair comparison, in each of the 3 TU datasets, we randomly select 20 graph samples that the victim model (we use the trained GCN models identical to those used in Section 4) originally classify correctly. For each of the sample, we generate $\{25, 50, 100\}$ perturbed samples that are 1 edit distance different from the original graphs and query the victim model to obtain their respective attack losses. We then train the surrogate models with the perturbed graphs and their attack losses as the training inputs and targets, and validate their performance on a further validation set of 200 perturbed graphs with the objective of predicting their attack losses. We use 3 metrics to evaluate the quality of the surrogate model: Root Mean Squared Error (RMSE) between predicted attack losses and the ground-truth atack losses, Spearman correlation between them and the negative log-likelihood on the validation set (which assess the quality of the prediction mean as long as its predictive uncertainty, as a principled uncertainty estimation is crucial in Bayesian optimisation). The results are shown in Fig. 10: it can be seen that the difference between Bayesian linear regression and GP models are insignificant in most cases in terms of RMSE and Spearman correlation (except in the PROTEINS dataset where GP model is arguably better), whereas the uncertainty estimation seems to be more stable throughout for Bayesian linear regression. It therefore justifies our usage of Bayesian linear regression, as its performance is often comparable with GP, but it is much more cheaper in terms of computation, making computations much more tractable especially when the number of queries is modestly large.

### D.2  Comparison with RL-S2V

We compare GRABNEL with RL-S2V on the graph classification dataset described in [10]. Each input graph is made of 1, 2 or 3 connected components. Each connected component is generated using the Erdős–Rényi random graph model (additional edges are added if the generated graph is disconnected). The label node features are set to a scalar value of 1 and the corresponding graph label is the number of connected components. The authors consider three variants of this dataset using different graph sizes, we consider the variant with the smallest graphs ($15 - 20$ nodes. The victim model, as well as the surrogate model used to compute Q-values in RL-S2V is structure2vec [9]. This embedding has a hyper-parameter determining the depth of the computational graph. We fix both to be the the smallest model considered in [10]. These choices were made to keep the computational budget to a minimum.

To adapt to the settings in [10], we only allow one edge edit (addition/deletion), and for GRABNEL we allow up to 100 queries to the victim model per sample in the validation set. For Random baseline, we instead allow up to 400 queries. Similar to [10], we enforce the constraint such that any edge edit must not result in a change of the number of disconnected components (i.e. the label) and any

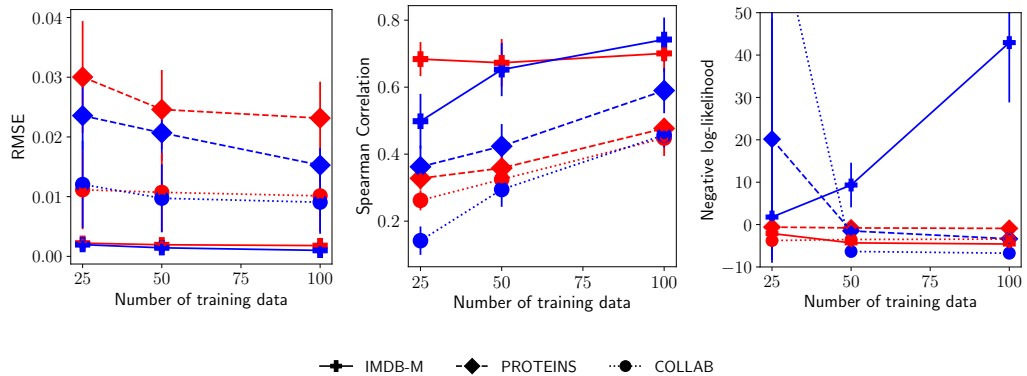

Figure 10: Comparison of the Bayesian linear regression and GP surrogate models on 3 TU datasets in terms of RMSE (lower is better), Spearman correlation (higher is better) and the negative log-likelihood (lower is better) on validation set. Error bars denote 1 standard error.

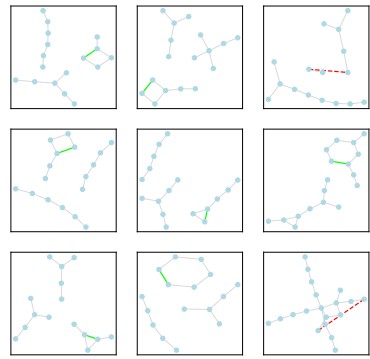

Figure 11: Adversarial examples found by the proposed method on the ER graphs with S2V being the victim model. Similar to Fig. 7, Red edges denote deleted edges from the original samples and green edges indicate those added.

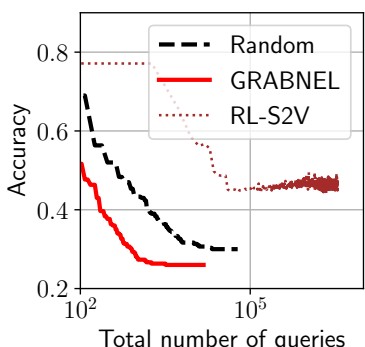

Figure 12: Validation accuracy vs *total* number of queries to the victim model. RL-S2V requires significantly more victim model queries, as it attempts to learn an attack policy by repeatedly querying a subset of the validation set which is used for policy training.

such edit proposed is rejected before querying the victim model. We show the results in Fig. 12, and we similarly visualise some of the adversarial samples found by GRABNEL in Fig. 11. The final performance of RL-S2V is similar to that reported in the [10], whereas we find that random perturbation is actually a very strong baseline if we give it sufficient query budget[5]. Again, we find that GRABNEL outperforms the baselines, offering orders-of-magnitude speedup compared to RL-S2V, with the main reasons being 1) GRABNEL is designed to be sample-efficient, and 2) GRABNEL does not require a separate training set *within* the validation to train a policy like what RL-S2V does. Fig. 11 shows that the edge addition is more common than deletion in the adversarial examples in this particular case, and often the attack agent forms *ring structures*. Such structures are rather uncommon in the original graphs generated from the Erdos-Renyi generator, and are thus might not be familiar to the classifier during training. This might explain why the victim model seems particularly vulnerable to such attacks.

### D.3 Additional Examples of Adversarial Samples Discovered

We show more examples of the adversarial examples found by GRABNEL on various datasets and victim models in Figs 13 and 14.

---

[5]The random baseline reported in [10] is obtained by only querying victim model with a randomly perturbed graph *once*.

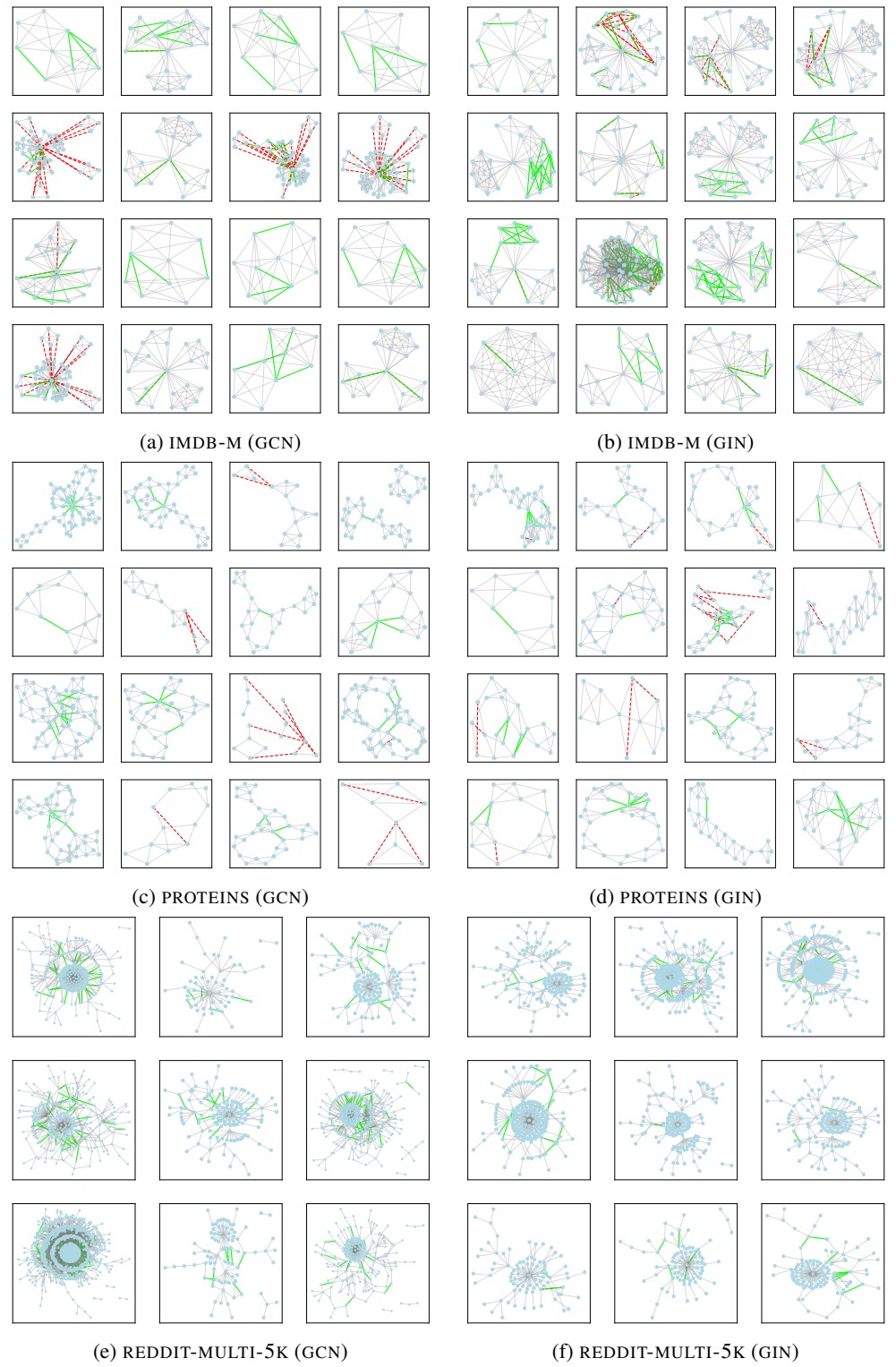

(a) IMDB-M (GCN)

(b) IMDB-M (GIN)

(c) PROTEINS (GCN)

(d) PROTEINS (GIN)

(e) REDDIT-MULTI-5K (GCN)

(f) REDDIT-MULTI-5K (GIN)

Figure 13: More examples of adversarial examples found by GRABNEL using GCN/GIN victim models. The colors have the same meaning as Fig. 7 in the main text.

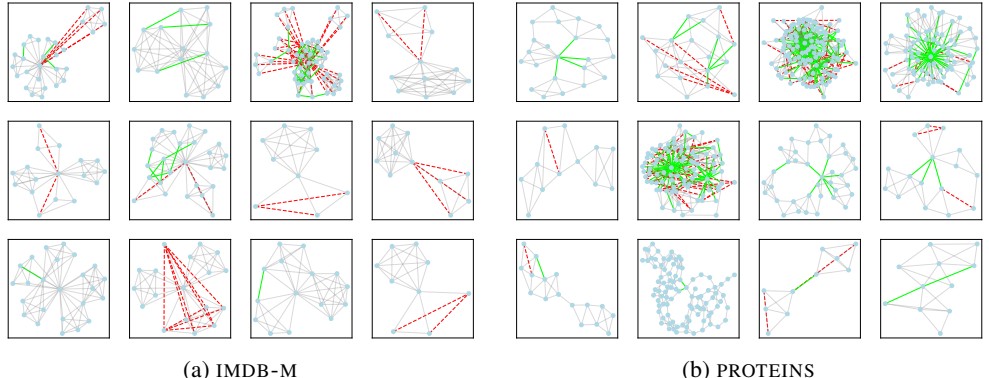

(a) IMDB-M            (b) PROTEINS

Figure 14: Examples of adversarial examples found by GRABNEL using Graph U-net victim models. The colors have the same meaning as Fig. 7 in the main text.

## D.4 REDDIT-MULTI-5K Results

For the REDDIT-MULTI-5K experiments, the graphs are in general typically much sparser and denoting the perturbation budget $\Delta$ with structural perturbation budget in terms of $n^2$ may be too lenient. Instead, we limit the perturbation budget in terms of the fraction of the *number of edges* of the individual graphs, and we set $\Delta \leq 0.03m$ for all experiments. We report the results in Table 5 and some examples of the adversarial examples discovered can be found in Fig. 13.

Table 5: Test accuracy of GCN and GIN victim models REDDIT-MULTI-5K before (*clean*) and after various attack methods.

|  | GCN [19] | GIN[45] |
|---|---|---|
| *Clean* | 45.20 | 48.40 |
| Random | 32.77 | 42.77 |
| Genetic [10] | 28.25 | 42.35 |
| GRABNEL (ours) | **23.73** | **29.27** |

## D.5 Ablation Studies

In this section, we conduct ablation studies on GRABNEL on two datasets previously considered in Section 4 in the main text, namely the PROTEINS dataset and the MNIST-75sp image classification task. We conduct the following variants of GRABNEL to understand how different components affect the performance:

- `Random` and `GA`: Identical to those in Sec. 4.
- `SequentialRandom`: instead of perturbing all edges simultaneously we use the sequential perturbation generation: we divide the total query budget into stages according to the description in Sec. 2, and at each stage we only modify one edge from the base graph *via random sampling* and commit to the perturbation that leads to the largest attack loss in the previous stage. This setup is otherwise identical to GRABNEL but the candidates are generated via random sampling instead of surrogate-guided BO.
- `GrabnelNoSequential`: GRABNEL but without the sequential perturbation selection. At each BO iteration, the BO needs to search and propose *all* (instead of one) edges to perturb up to the attack budget
- `Grabnel`: Full GRABNEL with both surrogate-guided BO and sequential perturbation selection.

We show the results in Fig. 15: it is evident that in both cases the use of surrogates and the use of sequential perturbation selection has led to improvements over baselines. In particular, `SequentialRandom` seems also to be a simple and strong baseline, with their final performance on par with GRABNEL but is less sample efficient. In PROTEINS, GRABNEL converges much faster initially (noting the log scale of the x-axis), while the final performance is comparable between

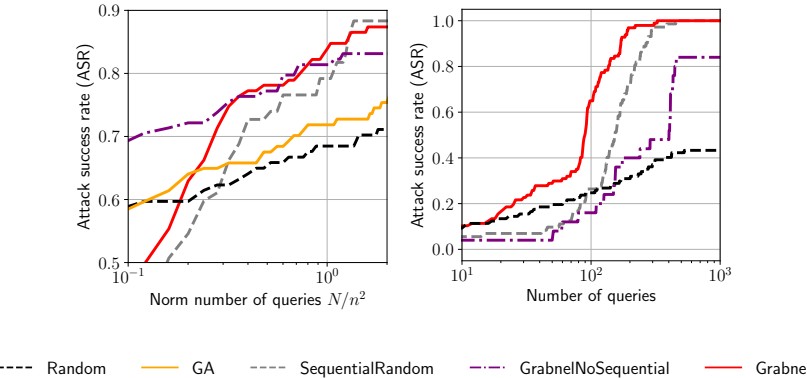

Figure 15: Ablation studies on PROTEINS and MNIST-75sp datasets.

GRABNEL and `SequentialRandom`. In the MNIST-75sp task, GRABNEL is roughly 1-2× faster throughout (it is worth noting that since we explicitly link the number of queries to the amount of perturbation applied, GRABNEL being 1-2× faster also suggests that it requires 1-2× less perturbation compared to `SequentialRandom` to reach the same ASR). The strong performance on PROTEINS of `SequentialRandom` is because 1) the perturbation budget $\Delta \leq 0.03n^2$ we set is relatively lenient: with many possible adversarial examples present in the search space, the importance of the different search algorithms may diminish (as we will show later, when we set a much stricter perturbation budget, the out-performance of GRABNEL is markedly larger. This could also be seen by the margin of out-performance in Fig. 15 when the number of queries is small), and 2) the dataset mainly features modestly-sized graphs on which simple a relatively small number random search might already be sufficient in finding vulnerable edges. On the other hand, `GrabnelNoSequential` improves significantly over `Random`, showing the effectiveness of the surrogate in BO and in PROTEINS it starts off with a much higher ASR because it utilises the entire attack budget throughout instead of the approaches using sequential perturbation selection, which is parsimonious with respect the amount of perturbation applied. Nonetheless, in both tasks its final performance is worse than the full GRABNEL, presumably due to the fact that the unconstrained search space, which scales exponentially with the attack budget (i.e. the number of edges to edit), is too large for the surrogate model to explore effectively even for modestly-sized graphs.

In view of the strong performance of `SequentialRandom`, we conduct more detailed experiments to compare it against the full GRABNEL and we show the results in Table 6: with exceptions of GCN on PROTEINS and Graph U-net on IMDB-M where the two perform similarly, possibly within margin of error, GRABNEL outperforms in all other cases: it is easy to see that the margin of GRABNEL over SequentialRandom increases significantly as the task difficulty increases. This is because SequentialRandom takes more queries to reach the similar level of ASR of GRABNEL and is less efficient in increasing the attack loss.

Table 6: Test accuracy of a GCN victim model after attacks by GRABNEL and `SequentialRandom` under perturbation budget $\Delta \leq 0.03n^2$. Mean (and ± standard deviation, if available) shown.

| Victim model | GCN | | | Graph U-Net | |
| Dataset | IMDB-M | PROTEINS | COLLAB | IMDB-M | PROTEINS |
|---|---|---|---|---|---|
| *Clean* | $50.53_{+1.4}$ | $71.23_{+2.6}$ | $79.93_{+2.1}$ | 55.33 | 79.46 |
| GRABNEL[*] | $\mathbf{45.23}_{+0.2}$ | $10.82_{+2.5}$ | $\mathbf{35.38}_{+9.3}$ | $\mathbf{41.33}$ | $\mathbf{58.80}$ |
| SequentialRandom | $46.22_{+0.2}$ | $\mathbf{10.12}_{+1.5}$ | $49.80_{+6.8}$ | 42.05 | 66.62 |

[*]: Taken from results in the main text.

However, as discussed, looking at the performance *only when a relatively lenient budget has been exhausted* can be misleading. We thus test the algorithms while allowing fewer edits, hence making the task more difficult. Taking the example on the PROTEINS dataset where `SequentialRandom` performs the strongest, we show a set of results with reduced perturbation budgets in Table 7 and the margin of GRABNEL over `SequentialRandom` increases significantly as the task difficulty increases. This is because `SequentialRandom` takes more queries to reach the similar level of ASR of

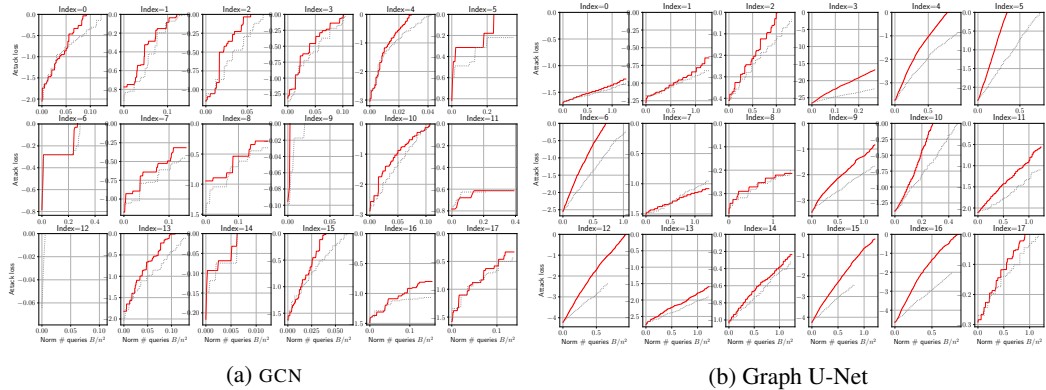

(a) GCN                 (b) Graph U-Net

Figure 16: Comparison of the attack loss as a function of the (normalised) number of queries to the GCN/Graph U-net victim models of GRABNEL and SequentialRandom on the PROTEINS dataset.

ASR of GRABNEL and is less efficient in increasing the attack loss: as concrete examples, we show the attack loss trajectory as a function of the number of queries to the victim model for both methods in randomly selected attack samples in Fig 16: in successfully attacked samples (i.e. the attack loss reaches 0), it is clear to see that GRABNEL typically requires fewer queries. Even for the samples that neither managed to successfully attack, GRABNEL pushes the losses closer to 0.

Table 7: Test acc. of GCN after attacks by GRABNEL and SequentialRandom on the PROTEINS dataset under varying perturbation budgets. Mean ($\pm$ standard deviation, if available) shown.

| $\Delta$ | $0.03n^2$ | $0.003n^2$ | $0.001n^2$ |
|---|---|---|---|
| GRABNEL | $10.82_{\pm2.5}$ | **26.43** | **52.59** |
| SequentialRandom | $\mathbf{10.12}_{\pm1.5}$ | 32.09 | 60.51 |

Another concrete example would be the attack on the MNIST-75sp task: while both GRABNEL and SequentialRandom converge eventually to 100% ASR, GRABNEL converges 2 times faster: on average, in a successfully attacked sample, GRABNEL rewires $1.84 \pm 0.7$ edges but SequentialRandom rewires $2.54 \pm 1.0$ edges (Fig 17), which suggests that SequentialRandom needs to modify 38% more edges on average to succeed. Imperceptibility in graph attack is usually measured in terms the number of edge edits, and thus while in some cases the end performance of SequentialRandom and GRABNEL can be similar especially when a lenient perturbation budget is given, GRABNEL is both more sample-efficient and produces less perceptible attacks. Therefore, we conclude that while SequentialRandom could perform strongly for easier tasks (e.g. high perturbation budget and smaller graphs) but otherwise GRABNEL is significantly better.

### D.6 Runtime Analysis

In this work, we particularly emphasise the desideratum of sample efficiency that we aim to find adversarial examples with the minimum number of queries to the victim model. We believe that this is a practical, and difficult, setup that accounts for the prohibitive monetary, logistic and/or opportunity costs of repeatedly querying a (possibly huge and complicated) real-life victim model. With a high query count, the attacker may also run a higher risk of getting detected. Given this objective, the cost of the algorithm should not only be considered from the viewpoint of computational runtime of the attack algorithm itself alone, and this is a primary reason why we use number of queries as the main cost criterion in our paper (this emphasis on the number of queries over runtime as the main cost metric is common in adversarial attacks in other data structures emphasising sample efficiency [28, 15, 49] and other related domains, like hyperparameter optimisation. The common

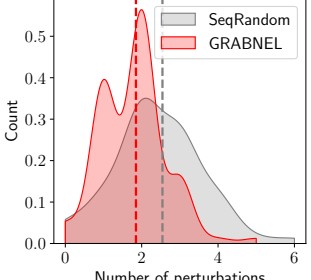

Figure 17: Distribution of the number of edge rewiring/swapping required to successfully attack MNIST-75sp samples in GRABNEL and SequentialRandom. The dotted lines denote the median numbers of edge operations required.

assumption is that the cost of the BO itself is secondary to the cost of querying the objective function (the cost here should not be interpreted as being the computing cost alone, but includes all the potential costs discussed above). Nevertheless, the runtime analysis is still a relevant metric, and we provide a more comprehensive analysis of the algorithm runtime below.

Each iteration in the main loop of our algorithm can be broadly separated into two parts:

1. Initialisation/updates of the surrogates: this step involves the WL feature extraction, and initialisation/update of the Bayesian Linear Regression (BLR) surrogate (the complexity of this step was discussed in Sec 2 with the new data. Note that BLR scales much better than GPs used in related works [28].

2. Acquisition function optimisation: this step involves using genetic algorithms to optimise the acquisition function. It is further broken into 2 sub-parts:

(a) GA steps, which involve the selection of the population and mutation and crossover operations on the parents. For the manipulations here, we do not have to store the full graphs, but we instead only have to maintain a tuple of the edges that are flipped/rewired.

(b) Conversion of the tuple into full graph objects (we use Deep Graph Library (DGL) [40] for implementation), and call the trained BLR to obtain predicted mean/variance and compute the acquisition function value.

Table 8: Runtime analysis of GRABNEL in terms of average second per iteration (standard deviation in brackets; slowest step in bold). $H$ denotes the number of WL iterations performed. Benchmarked on an otherwise idle machine with AMD Ryzen 7 CPU and 32 GB RAM. We used GCN victim model. Results may vary significantly depending on the hardware, system load and the hyperparameters used

| # nodes | # edges | $H$ | Step 1 | Step 2a | Step 2b |
|---------|---------|-----|--------|---------|---------|
| 17 | 106 | 1 | 0.022 (0.0027) | 0.0344 (0.0002) | **0.482 (0.006)** |
| 72 | 719 | 1 | 0.251 (0.003) | 0.0371 (0.0006) | **0.458 (0.009)** |
| 1961 | 5336 | 0 | **1.76 (0.12)** | 0.0555 (0.0007) | 1.52 (0.016) |

Note that even for a graph with almost 2k nodes and >5k edges (which is larger than most graphs in the TU dataset for graph classification), the runtime is still manageable on a mainstream desktop-grade PC. In fact, the GA itself is efficient, and the much slower step is Step 2b: instead of our algorithm itself being inefficient, this is because of the large overhead in graph representation conversion between the list of changed edges (e.g. $([1, 2], [3, 4])$, which denotes a perturbed graph with edges $e_{(1,2)}$ and $e_{(3,4)}$ flipped from the original graph) and an actual DGL graph object. For better efficiency, it should be possible to reduce the number of such conversions as the perturbed and original graphs differ only at the flipped edges which only make up a very small fraction of all edges. At the very least, since each conversion is independent for other graphs, we can parallelise it for candidates in the population of the genetic algorithm (the current code does this sequentially).

To give a better context, on a shared Intel Xeon Gold server where we conduct the majority of experiments (we unfortunately could not provide very reliable and accurate statistics due to the varying load by other users), each attack on a single graph on the IMDB-M (average 66 edges per graph) dataset usually takes < 1min. On COLLAB, which on average is much larger, each attack on average takes 1h (Note we set the maximum number of queries to be dependent on the sizes of the graph, so larger graphs are also proportionally allocated a higher query budget and hence each run could be much longer). Furthermore, the sizes of graphs within a dataset can vary a lot, and runtime also depends a lot on the difficulty of attack (if an adversarial perturbation is easily found the run is terminated early). A further comparison with RL-S2V [10] is that to run the same task on ER-graphs attack with 15-20 nodes, our method takes 30min - 1h. RL-S2V, which requires a separate validation set to train policy on, requires approx. 1.5h with GPU acceleration and approx. 12h without (we do not currently use any GPU acceleration for our method).