# OpenReview forum: "Adversarial Attacks on Graph Classifiers via Bayesian Optimisation"
_NeurIPS.cc/2021/Conference — NeurIPS 2021 Poster_

### Official Review · Reviewer_PcFP · 2021-07-04

**Rating:** 5
**Confidence:** 5

**Summary:**

The paper studies adversarial attacks to graph neural networks for graph classification. The authors design a Bayesian optimization-based attack that is black-box, query-efficient, and parsimonious with respect to the perturbation. The proposed attack is evaluated on three benchmark graph datasets and shows its effectiveness.

Strengths
+ The studied problem is important
+ The proposed attack is evaluated on multiple application domains
+ Attack analysis is interesting

Weaknesses
- Novelty is unclear
- The proposed attack is hard to follow
- Evaluation is insufficient
- Missing important references
- No discussion on defense

**Limitations And Societal Impact:**

The authors addressed the limitations and potential negative societal impact of their work

**Main Review:**

The flowchart of the proposed bayesian optimization-based attack is unclear to me.  I suggest the authors add an overview to high-level show the motivation and the attack flowchart.

As I do not fully understand the motivation and details of the attack, I also cannot quite catch the key contributions. For example, why the proposed bayesian optimization is suitable for attacking GNNs?

Why GRABNEL outperforms white-box gradient-based attack? Why gradient-based attack performs poorly on PROTEINS with GCN, while performing the best on the other datasets?
GRABNEL and random attack obtain comparable attack performance against fake news detector. This may reveal that GRABNEL is not effective enough.

Lack of comparison with (2) and (3)


The authors should cite and discuss the following references on adversarial attacks to graphs/graph neural networks


(1) Attacking Graph-based Classification via Manipulating the Graph Structure

(2) Black-box adversarial attacks on graph neural networks with limited node access

(3) Node injection attacks on graphs via reinforcement learning

(4) Indirect Adversarial Attacks via Poisoning Neighbors for Graph Convolutional Networks

(5) Evasion Attacks to Graph Neural Networks via Influence Function

(6) Adversarial examples on graph data: Deep insights into attack and defense

(7) Topology attack and defense for graph neural networks: An optimization perspective


The authors also lack of discussion of defenses against adversarial attacks to graph neural networks.


Other comments:
[17] is not a reinforcement learning-based techniques

a appropriate surrogate => an appropriate surrogate

“we note that modifying ...., which correspond to modifying existing accounts  and tweets, is considered impractical and prohibited.” => I do not think this claim is correct.

Suggestions:
-Add an overview to well motivate the proposed attack
-Make the contributions more clear
-Compare with the existing attacks
-Discussion on defenses



**Time Spent Reviewing:**

4

---

> ### Author Response · Authors · 2021-08-10
> **Thank you for your comments**
>
> We thank the reviewer for their insightful comments. Please see below for our detailed response.
>
> 1. >*The flowchart of the proposed bayesian optimization-based attack is unclear to me. I suggest the authors add an overview to high-level show the motivation and the attack flowchart.*
>
> We thank the reviewer for the suggestion and a flow chart will be added to the revised paper.
>
> To describe the procedure in words, we adopt a query-based iterative process to modify a target graph to achieve evasion attack. At each instance, with a number of candidate graphs, we first use WL extractor to decompose the graph into vector representation, and use sparse Bayesian linear regression based Bayesian optimization to select the most promising candidate to query the victim model. We repeat the process until either the attack succeeds, or the budget is exhausted.
>
> 2. >*As I do not fully understand the motivation and details of the attack, I also cannot quite catch the key contributions. For example, why the proposed bayesian optimization is suitable for attacking GNNs?*
>
> On the question on contribution and motivation, the reviewer is referred to our general response. The general motivations are also in the paper in the paragraph starting from Line 35.
>
> BO is suitable as the attack problem is black-box, and we aim to be query efficient; both desiderata are fulfilled by BO. While to our knowledge, we first apply BO to graph settings (which is more difficult due to the discrete nature of the data structure), there are existing works that apply BO in query-efficient image adversarial attacks [Ru 2019, Zhang 2021]. We believe these works could also be helpful in understanding the motivation of using BO in our context.
>
> References:
>
>  Ru, B., Cobb, A., Blaas, A. and Gal, Y., 2019, September. Bayesopt adversarial attack. In International Conference on Learning Representations.
>
>  Zhang, Z. and Yu, S., 2021. BO-DBA: Query-Efficient Decision-Based Adversarial Attacks via Bayesian Optimization. arXiv preprint arXiv:2106.02732.
>
> 3. >*Why GRABNEL outperforms white-box gradient-based attack? Why gradient-based attack performs poorly on PROTEINS with GCN, while performing the best on the other datasets?   GRABNEL and random attack obtain comparable attack performance against fake news detector. This may reveal that GRABNEL is not effective enough.*
>
> While the gradient-based method often performs strongly because it is white-box, there is no guarantee that it will always perform strongly. There are two main possible reasons:
> 1. For general edge flipping problems where we may create edges, we have to compute gradients w.r.t. all possible edges (including those that do not currently exist). This gradient can be highly inaccurate.
> 2. Gradients only capture local information and they are not necessarily accurate when used to extrapolate function value beyond that neighbourhood. However, relying on gradients to select edge perturbations constitutes such an extrapolation, as edge addition/deletion is binary (e.g. denoting edge absent=0 and edge present=1, when we are selecting an edge to create, we are using the gradients at 0 to extrapolate the loss value at 1. Vice versa if we are deleting an edge).
>
> These two points mean that gradients may not only be inaccurate but sometimes even be misleading, which could account for their poor performance sometimes.
>
> On fake news detection, we note that the graphs in consideration are relatively small. In these settings with moderate search space, random search is competitive (we can just randomly enumerate all possibilities. If an adversarial example indeed exists, it will be eventually picked up) yet GRABNEL still outperforms. Furthermore, in all other experiments GRABNEL outperforms significantly especially when the search space is larger, so we do not agree with the statement that “GRABNEL is not effective enough”.
>
> 4. >*Comparison with existing works mentioned by the reviewer*
>
> Firstly, we would like to emphasise that **none** of (1) - (7) referred to by the reviewer actually attacks graph classification, even though the *task* of graph classification is mentioned by multiple papers (e.g: (2) and (5)). This exactly shows that there is a gap in the literature to be filled, as the community is well-aware of the importance of graph classification, but there are few existing works studying the adversarial attack against it.
>
> Other than that, (3) and (4) are poisoning attacks but we are focusing on evasion attacks.
>
> (6) and (7) both utilise model gradients and hence are white-box. Note that we require no access to the model parameters/gradients.
>  While (1) is compatible with a similarly restrictive setup (Parameter=No, Training=No, Graph=Partial. Sec 4.2), they require substitute parameters & training dataset to perform attack. At the inference time of graph classification, we have no access to any training inputs at all (we may not even have a test “set”, if, say, we are asked to attack on a single graph. This is different from node classification where training and testing take place on a single graph), so it is unclear how to use substitute parameters/training dataset in this context.
>
> (5) uses node label influence, but in the graph classification case we do not even have node labels (we only have a label for the entire graph) so generalising the method to graph classification is at least non-trivial.
>
> Finally, (2) is black-box evasion attack for node classification. At least in the current paper, they focus on the problem of selecting a subset of nodes to attack (so naturally not at graph level), it also seems to only attack node features (by adding noise to node features) and not topology. In contrast, our method handles both topology and features (in MNIST-75sp the surrogate model handles both node and edge features). While our present work mainly focuses on topology attack, we may easily adapt GRABNEL for feature-level/hybrid attack. In fact, feature level attack could be an easier task if the perturbation is continuous, as we could possibly rely on existing gradient-based acquisition optimization of BO which is well-suited for continuous domain.
>
> 5. > *The authors also lack of discussion of defenses against adversarial attacks to graph neural networks.*
>
> Similar to our response to Reviewer FwKT, we would like to emphasise that literature on adversarial attack on graph classification itself is very scarce (not to mention defence). To develop possible defences, we believe the first step would be to understand the attack patterns (which is not done yet in graph classification -- it is also worth noting that these graph-level patterns are not necessarily the same as node-level patterns studied in previous work. See Sec 5), and is a reason why we include analyses in Sec 5. We hope these insights could lead to better defense strategies, which is a very interesting direction to explore especially due to the scarcity of literature in attacking graph classification.
>
> 6. >*[17] is not a reinforcement learning-based techniques*
>
> We believe [17] is a reinforcement learning technique. Excerpt from the abstract of [17]:
>
> *…In this paper, we propose a graph rewiring operation which affects the graph in a less noticeable way compared to existing operators. **We then use reinforcement learning to learn the attack strategy based on the proposed rewiring operation**. Experiments on real world graphs demonstrate the effectiveness of the proposed framework...*
>
> 7. >*“we note that modifying ...., which correspond to modifying existing accounts and tweets, is considered impractical and prohibited.” => I do not think this claim is correct.*
>
> In our setting nodes correspond to users and edges correspond to tweets. We typically cannot modify existing node features (properties of existing users) or edges (modifying existing tweets) unless we can control these existing Twitter accounts, which is highly unrealistic. The much easier action one can potentially do is to add new twitter accounts in the conversation (new node) and retweet/reply to existing tweets (new edges). We would be grateful if the Reviewer could elaborate more on why they think this claim is incorrect.

---

> > ### Comment · Reviewer_PcFP · 2021-08-25
> > **Adversarial Attacks on Graph Classifiers via Bayesian Optimisation**
> >
> > Thanks for your response. I raised my score to "Marginally below the acceptance threshold".

---

### Official Review · Reviewer_Jxr4 · 2021-07-15

**Rating:** 2
**Confidence:** 5

**Summary:**

A very similar version has been accepted into the ICML 2021 Workshop, probably from the same authors. See the link https://openreview.net/forum?id=7oziDfK4Fs for details. Hence, I have to reject this paper, due to the duplicate submission.

**Ethical Concerns:**

A very similar version has been accepted into the ICML 2021 Workshop, probably from the same authors. See the link https://openreview.net/forum?id=7oziDfK4Fs for details. Hence, I have to reject this paper, due to the duplicate submission.

**Ethics Review Area:**

["I don’t know"]

**Limitations And Societal Impact:**

A very similar version has been accepted into the ICML 2021 Workshop, probably from the same authors. See the link https://openreview.net/forum?id=7oziDfK4Fs for details. Hence, I have to reject this paper, due to the duplicate submission.

**Main Review:**

A very similar version has been accepted into the ICML 2021 Workshop, probably from the same authors. See the link https://openreview.net/forum?id=7oziDfK4Fs for details. Hence, I have to reject this paper, due to the duplicate submission.

**Time Spent Reviewing:**

45 mins

---

> ### Author Response · Authors · 2021-08-10
> **We did not violate the double submission policy**
>
> We would like to clarify that we did not violate the double submission policy.
>
> Firstly, we will not comment on the speculation of possible author identities in the review due to the double-blind policy. We'd hope the reviewer could remove the external link that could potentially deanonymize the process.
>
> Secondly, even if a highly similar version of the paper was indeed submitted to the said workshop, there is no violation of the double submission policy in either NeurIPS or the workshop:
>
> Excerpt from NeurIPS FAQ https://neurips.cc/Conferences/2021/PaperInformation/NeurIPS-FAQ
>
> >*Can I submit work to NeurIPS and then later submit the same work to a non-archival venue while it is still under review at NeurIPS? Yes, as long as this does not violate the other venue's policy on dual submissions (if it has one).*
>
> Excerpt from the ICML AdvML website the reviewer referred to (https://advml-workshop.github.io/icml2021/):
>
> >*We only consider submissions that haven’t been published in any peer-reviewed venue, including ICML 2021 conference. We welcome submissions that are currently under review in some conferences (e.g., NeurIPS 2021) The workshop is non-archival and will not have any official proceedings.*

---

### Official Review · Reviewer_FwkT · 2021-07-16

**Rating:** 5
**Confidence:** 4

**Summary:**

The paper addresses the task of designing adversarial attacks that perturb the structure of a graph to induce errors in graph classification. The approach is based on Bayesian optimization – the authors employ a surrogate model that couples a WL feature extractor with sparse Bayesian linear regression to learn a model of the attack loss. The algorithm operates in the black-box evasion setting, and aims to limit the number of queries to the model. A genetic algorithm is employed to derive graph proposals which are subsequently evaluated using the learned surrogate model. Numerical experiments demonstrate that the proposed method outperforms the best existing methods and significantly reduces the number of required model queries

**Ethical Concerns:**

There are no ethical issues.

**Limitations And Societal Impact:**

The authors have addressed  the limitations and potential negative societal impact of their work.

**Main Review:**

The paper is well-written and for the most part, the algorithm is clearly described.

It is always difficult to manage space constraints, but I thought there could have been more information included in the actual paper about the WL feature extractor. The paper also seems to rush through the Bayesian optimization component of the algorithm (for example, the use of EI is mentioned only parenthetically and there is no justification of this choice). Since this is a focus of the contribution, I feel more description could have been allocated to it. For example, considerable space is devoted to the genetic algorithm and this seems to have been borrowed directly from [7] – the authors don’t highlight the modifications they have made.

Strengths:

(1) The paper provides a novel adversarial attack procedure. The approach is well-motivated and clearly described and there is an innovative combination of methods to improve performance.

(2) Numerical experiments demonstrate that the developed approach outperforms existing methods for some victim models.

Weaknesses:

(1) Novelty: Overall I believe there is sufficient novelty to warrant acceptance. To the best of my knowledge this is the first application of Bayesian optimization for adversarial attacks on graph data, so that is a positive. On the other hand, most of the ingredients used in the algorithm are relatively minor adaptations of previous work. For example, the Bayesian optimization to search over graphs in [23] is closely related to the core of the technique presented in this paper, and while the adaptation from (Gaussian process + WL kernel) to (WL feature extractor plus linear regression) is important to reduce computation time, it is not a major innovation. The algorithm also employs (a variant) of the genetic algorithm from [7] to generate graph proposals. So the paper consists of combinations and adaptations of existing methods to address a problem that has not received a significant amount of attention.

(2) The paper only considers GCN and GIN as the victim models (and ChebyGIN). These are far from the state-of-the-art in terms of graph classification. The authors do not really explain why they have only explored these graph models. It leads to concerns that the results may not carry over to the improved methods.

(3) The authors do not examine the efficacy of the approach for settings where robust GNNs are used (eg. [R1], but there are multiple others). Many of these robust GNNs involve training over perturbations of the observed graphs. This may well substantially reduce how effective the proposed method is.

(4) The authors do not discuss or explore the possibility of some form of adversarial defence. Although most of the proposed defences do not target the black-box, evasion setting, something like the GCN-Jaccard defence in [R2], which just pre-processes the graph, removing dubious edges, could easily be adapted to the setting.

In general, when reading a paper that proposes attacks, one expects to see (i) an investigation of the efficacy with respect to multiple victim models; (ii) a discussion and exploration of the effectiveness against robust models; (iii) a discussion and exploration of the ability to navigate existing defences (or if they do not exist, baseline defences that are simple to propose).

[R1] Wei Jin, Yao Ma, Xiaorui Liu, Xianfeng Tang, Suhang Wang, and Jiliang Tang. 2020. Graph Structure Learning for Robust Graph Neural Networks. In Proc. ACM SIGKDD International Conference on Knowledge Discovery & Data Mining (KDD '20).

[R2] Huijun Wu, Chen Wang, Yuriy Tyshetskiy, Andrew Docherty, Kai Lu, and Liming Zhu. Adversarial examples for graph data: Deep insights into attack and defense. In International Joint Conference on Artificial Intelligence, IJCAI, pp. 4816–4823, 2019.


**Time Spent Reviewing:**

4

---

> ### Author Response · Authors · 2021-08-10
> **Thank you for your comments**
>
> We thank the Reviewer for their appreciation of and insightful comments on our work. Please see below for our detailed response
>
> 1. >*Overall I believe there is sufficient novelty to warrant acceptance. To the best of my knowledge this is the first application of Bayesian optimization for adversarial attacks on graph data, so that is a positive.*
>
>    >*On the other hand, most of the ingredients used in the algorithm are relatively minor adaptations of previous work... So the paper consists of combinations and adaptations of existing methods to address a problem that has not received a significant amount of attention.*
>
> We thank the reviewer for acknowledging the novelty of our paper. Although many components of the algorithm are known, we would like to emphasise that our method is an innovative combination of these techniques to address an important and difficult problem.
>
> On specific notes, it is worth mentioning that although our linear regression + WL extractor is indeed conceptually similar to the GPWL used in [23], the tasks in question are significantly different. In NAS, we typically have small directed acyclic graphs with only discrete node/edge features (representing the different operations). On the other hand, the surrogate in our paper deals with a much richer variety of graphs, such as those significantly larger and/or with continuous attributes. These all require a non-trivial amount of thinking to design an appropriate surrogate to handle the additional challenges.
>
>
> 2. >*The paper only considers GCN and GIN as the victim models (and ChebyGIN). These are far from the state-of-the-art in terms of graph classification. The authors do not really explain why they have only explored these graph models. It leads to concerns that the results may not carry over to the improved methods.*
>
> We select GCN and GIN as these are the most popular victim models considered in literature (for example, refer to the survey paper [Sun 2020]), so we believe they are good models to start with. These are also one of the most generic models, and many strong models are modified from/inspired by them.
>
> We select ChebyGIN on MNIST-75sp dataset as it is reported to be the highest performing model in the paper that originally proposed MNIST-75sp and we directly use the pretrained model released by the authors.
>
> We agree that looking at further, potentially better performing SoTA architectures (e.g. DiffPool, Graph UNets) is an important future direction.  Unfortunately due to the time and compute constraint, we have not been able to fully evaluate our methods and baselines on these SotA models during the initial response period. However, since attack is formulated as a black-box, model-agnostic optimisation problem, there is no obvious reason why our method would not work. With that said, we do look forward to adding results on these additional models in the revised paper.
>
> References:
> [Sun et al]: https://arxiv.org/abs/1812.10528
>
> 3. >*The authors do not examine the efficacy of the approach for settings where robust GNNs are used (eg. [R1], but there are multiple others). Many of these robust GNNs involve training over perturbations of the observed graphs. This may well substantially reduce how effective the proposed method is*
>
> .We would like to note that both methods are for node classifications and transferring them to graph classification is not necessarily trivial. For example, in light of Sec 5, it is worth noting that some adversarial patterns (such as the observations on how the attack changes community structures) are graph-level, which is not necessarily consistent with node-level observations, and thus whether node-level defence generalises is an open question.
>
> To develop possible defences, we believe the first step would be to understand the attack patterns (which is not done yet in graph classification), and is a reason why we include analyses in Sec 5. We hope these insights could lead to better defense strategies, which is a very interesting direction to explore especially due to the scarcity of literature in attacking graph classification.
>
>
> 4.  >*In general, when reading a paper that proposes attacks, one expects to see (i) an investigation of the efficacy with respect to multiple victim models; (ii) a discussion and exploration of the effectiveness against robust models; (iii) a discussion and exploration of the ability to navigate existing defences (or if they do not exist, baseline defences that are simple to propose)*
>
> In principle, while we agree with the reviewer on the points to be expected in an adversarial attack paper, we emphasise that the scarcity of previous works has made some aspects difficult to implement. Specifically, on (i) we do provide an investigation wrt multiple victim models, although we agree there is scope to include more. On (ii) and (iii), however, we would like to emphasise that as the reviewer agrees, graph-level adversarial attack itself is not as well-developed as the other domains, and in light of defence, we believe the analysis in this paper is a first step towards this direction already.

---

> > ### Comment · Reviewer_FwkT · 2021-08-25
> > **Comments on response**
> >
> > Thank you for the response. The responses to all reviewers are very thorough, and I appreciate the time taken by the authors to provide the detailed feedback.
> >
> > Given the limited time to provide a response, I think it was challenging to address all of the concerns I raised, especially since most revolved around a request for more experimental results.
> >
> > With regard to victim models, I agree that GCN and GIN are, of course, very important to include, but I really think it is essential to include at least one or two of the state-of-the-art models in the experiments. I am not sure I completely agree with the claim "However, since attack is formulated as a black-box, model-agnostic optimisation problem, there is no obvious reason why our method would not work".  I agree that in all likelihood the approach "would work", but many of the more recent techniques include training with perturbations to the graph structure, and I don't think one can say with confidence, without experimental support, that similar levels of performance improvement will automatically carry over when more recent models are used as victims.
> >
> > (2) When proposing an attack, I do think it is essential that there are some experiments examining (a) robust GNNs and (b) detection and defence strategies.
> >
> > I should indeed have cited papers that address graph classification in my review. Having said this, there are several relevant papers that do directly address graph classification that should have been discovered during the literature review. For example, You et al. "Graph Contrastive Learning with Augmentations", NeurIPs 2020, contains graph classification results (as its main experiments) and explores the robustness to attacks. Chen et al., “Adversarial Detection on Graph Structured Data” (PPMLP 2020) proposes a method for detecting attacks for graph classification; Zhang et al. “Backdoor Attacks to Graph Neural Networks” (arXiv 2020, SACMAT 2021) propose a simple defence method (randomized smoothing); Gao et al. “Certified Robustness of Graph Classification against Topology Attack with Randomized Smoothing” (Globecom 2020) also employ randomized smoothing.
> >
> > I think the main idea of the paper is elegant and with more extensive experimentation, I would readily recommend acceptance. But for the moment I think there needs to be some more work before it is ready for publication.

---

### Official Review · Reviewer_8giU · 2021-07-28

**Rating:** 6
**Confidence:** 4

**Summary:**

In this paper, the authors propose GRABNEL, a black-box adversarial attack against graph-level classifiers. The intuition is that by leveraging Bayesian Optimization(BO) techniques, one can reduce the number of queries needed. The authors propose to use a Weisfeiler-Lehman (WL) feature extractor to extract vector representations of the input graphs. After that, a Sparse Bayesian linear regression surrogate model is used to approximate the adversarial loss. The query budget is divided into different stages. For each stage, the attacker modifies the graphs and queries the target model before it runs out of query budget. The modified graphs, as well as the observed predictions, are fed into a genetic algorithm to generate the perturbation for the current stage. After that, the attacker queries the target model and updates the surrogate model accordingly. Experiments show that the proposed approach achieves a higher success rate than the baselines.

**Limitations And Societal Impact:**

## Weaknesses
- The details of the proposed approach is not clear, see the comments for more information
- The efficiency of the proposed approach is unclear

**Main Review:**

## Strengths
- The idea of introducing BO to the graph adversarial attacks is interesting
- The authors provide a detailed analysis

## Comments

The proposed approach achieves a higher success rate than the baselines and the idea of introducing BO to the graph adversarial attacks is interesting. However, there're some concerns about the paper.

Sparse Bayesian linear regression models are used to estimate the adversarial loss of the target model. However, the details of these models are unclear. For example, what's the architecture of these models? How do they interact with other parts of the algorithm? The paper will be more concrete if the authors could provide these details. Besides, I think it is better to introduce Bayesian Optimization as background knowledge, as it will make the paper more comprehensive

Instead of training a separate shadow model, the proposed approach uses a Sparse Bayesian linear regression surrogate model to approximate the adversarial loss. Since the approximation is done for a particular input sample, this process has to be repeated if the attacker wishes to generate more adversarial samples. This leads to the question that if the proposed approach is computationally efficient. It will be better if the authors could elaborate on this.

Some inconsistencies found in the paper:
- No reference to Fig.6 is found.
- 'implementation details... are presented in App. C' (Line 211), which should be in App. D
- 'In fact, as we elaborate in Sec. 6 ..'(Line 247), which should be in Sec.5

**Time Spent Reviewing:**

4

---

> ### Author Response · Authors · 2021-08-10
> **Thank you for your comments**
>
> We thank the reviewer for the positive and insightful feedback, and please see below for our response to your concerns:
>
> > *Sparse Bayesian linear regression models are used to estimate the adversarial loss of the target model. However, the details of these models are unclear... Besides, I think it is better to introduce Bayesian Optimization as background knowledge...*
>
> We thank the reviewer for the suggestion and we will include BO as background knowledge.
> Sparse Bayesian linear regression on itself is not different from classical linear regression in terms of its input and targets, but differs in the fact that Bayesian linear regression also outputs an uncertainty estimate. The WL feature extractor (Sec 2) produces a vector representation of the graphs, which are then passed to the Bayesian linear regression.
> We will add a system-level diagram to show the overall interactions of the different components of the algorithm.
>
> >*Instead of training a separate shadow model, the proposed approach uses a Sparse Bayesian linear regression surrogate model to approximate the adversarial loss. Since the approximation is done for a particular input sample, this process has to be repeated if the attacker wishes to generate more adversarial samples. This leads to the question that if the proposed approach is computationally efficient.*
>
> The setup we consider is *sample-efficient* black-box attack on graph classification with minimal amount of perturbation. We believe that this is a practical, and difficult, setup that accounts for the prohibitive monetary, logistic and/or opportunity costs of repeatedly querying a (possibly huge and complicated) real-life victim model. With a high query count, the attacker may also run a higher risk of getting detected.
> Given this objective, the cost of the algorithm should *not* only be considered from the viewpoint of computational runtime of the attack algorithm itself alone (unlike several existing works such as [1, 2] where sample efficiency is not emphasised), and this is a primary reason why we use number of queries as the main cost criterion in our paper. It is worth noting that this emphasis on the number of queries over runtime as the main cost metric is also common in adversarial attacks in other data structures emphasising sample efficiency ([3, 4, 5]). The common assumption is that the cost of the BO itself is secondary to the cost of querying the objective function (the cost here should not be interpreted as being the computing cost alone, but includes all the potential costs discussed above). Admittedly, the overhead cost of BO can be higher than competing methods, such as those based on random search, because BO maintains a statistical surrogate model and runs an inner acquisition optimisation to actively and carefully select each sample location, trading off slight increase in computation overhead for bigger improvement in sample efficiency.
> With that said, we would also like to provide a more comprehensive analysis of the algorithm runtime:
>
> Each iteration in the main loop of our algorithm can be broadly separated into two parts:
>
> 1. Initialization/updates of the surrogates: this step involves the WL feature extraction, and initialization/update of the Bayesian linear regression (BLR) surrogate (the complexity of this step was discussed in Sec 2) with the new data. Note that BLR scales much better than GPs used in related works [3].
>
> 2. Acquisition function optimization: this step involves using genetic algorithms to optimize the acquisition function. It is further broken into 2 sub-parts:
>
>       a) genetic algorithm steps, which involve the selection of the population and mutation and crossover operations on the parents. For the manipulations here, we do not have to store the full graphs, but we instead only have to maintain a tuple of the edges that are flipped/rewired.
>
>       b) convert the tuple into full graph objects (we use Deep Graph Library (DGL) for implementation), and call the trained BLR to obtain predicted mean/variance and compute the acquisition function value.
>
> Here we provide some quantitative results of our algorithm running on graphs of different sizes:
>
> Table R1: Runtime analysis of our algorithm in terms of average sec/iter (standard deviation in brackets. Slowest step bolded). Benchmarked on a machine with AMD Ryzen 7 3700X CPU and 32 GB 2666MHz RAM. Victim model = GCN. Results may vary depending on the hardware, system load and hyperparameters.
>
>
> | \# nodes | \# edges |Step 1 | Step 2a  | Step 2b |
> | :--- | :--- | :--- | :--- | :--- |
> | 17|106|0.022 (0.0027)|0.0344 (0.0002)|**0.482 (0.006)**|
> |72| 719|0.251 (0.003)|0.0371 (0.0006)|**0.458 (0.009)**|
> | 1961| 5336|**1.76 (0.12)** |0.0555 (0.0007) | 1.52 (0.016)|
> |||||
>
> Note that even for a graph with almost 2k nodes and >5k edges (which is larger than most graphs in the TU dataset for graph classification), the runtime is still manageable on a mainstream desktop-grade PC. In fact, the genetic algorithm itself is efficient, and the much slower step is Step 2b: instead of our algorithm itself being inefficient, this is because of the large overhead in graph representation conversion between the list of changed edges (e.g. $([1,2], [3,4])$, which denotes a perturbed graph with edges $e_{(1, 2)}$ and $e_{(3, 4)}$ flipped from the original graph) and an actual DGL graph object. For better efficiency, it should be possible to reduce the number of such conversions as the perturbed and original graphs differ only at the flipped edges which only make up a very small fraction of all edges. At the very least, since each conversion is independent for other graphs, we can parallelise it for candidates in the population of the genetic algorithm (the current code does this sequentially).
>
> To give a better context, on a shared Intel Xeon Gold server (we unfortunately could not provide very reliable and accurate statistics due to the varying load by other users), each attack on a single graph on the IMDB-M (average ~66 edges per graph) dataset usually takes < 1min. On COLLAB, which on average is much larger (average ~2500 edges per graph), each attack on average takes up to ~1h (up to 20,000 queries and around 4,000 iterations assuming batch size of 5). Note we set the maximum number of queries to be dependent on the sizes of the graph, so larger graphs are also proportionally allocated a higher query budget and hence each run is much longer. Furthermore, the sizes of graphs within a dataset can vary a lot, and runtime also depends a lot on the difficulty of attack (if an adversarial perturbation is easily found the run is terminated early). A further comparison with RL-S2V is that to run the same task on ER-graphs attack with 15-20 nodes, our method takes 30min ~ 1h. RL-S2V, which requires a separate validation set to train policy on, requires ~1.5h with GPU acceleration and ~12h without (we don’t use any GPU acceleration throughout for our method).
>
> On the point of shadow model, we note that it’s much more difficult to train separate shadow models in the graph classification setup without additional data.  In a semi-supervised node classification setup, we can see all the nodes and labels of training nodes at all times as training and testing take place on the same graph (which could be used to train shadow models, such as in Nettack). In our graph classification setup, however, at test time we have no access to training input/labels. While certain existing methods like RL-S2V try to train a policy so that we only need to run the algorithm once (instead of per-sample), they need a separate validation set (that is representative enough of the test set) on which they repeatedly query and train a policy. In a practical setting this is rarely possible and even on benchmarks it is difficult to obtain a small subset of data that is also representative enough, as the graphs can vary significantly even within the same dataset.
>
> References:
>
> [1] Dai, H., Li, H., Tian, T., Huang, X., Wang, L., Zhu, J. and Song, L., 2018, July. Adversarial attack on graph structured data. In International conference on machine learning (pp. 1115-1124). PMLR.
>
> [2] Ma, Y., Wang, S., Derr, T., Wu, L. and Tang, J., 2019. Attacking graph convolutional networks via rewiring. arXiv preprint arXiv:1906.03750.
>
> [3] Ru, B., Cobb, A., Blaas, A. and Gal, Y., 2019, September. Bayesopt adversarial attack. In International Conference on Learning Representations.
>
> [4] Huang, Z., Huang, Y. and Zhang, T., 2020. CorrAttack: Black-box Adversarial Attack with Structured Search. arXiv preprint arXiv:2010.01250.
> .
>
> > *No reference to Fig.6 is found and Minor typos*
>
> We thank the reviewer for spotting these, and these errors will be redressed in the revised version of the paper.

---

### Author Response · Authors · 2021-08-10
**Overall response to all reviewers**

We thank the reviewers for their insightful feedback. In this general response, we would like to reiterate the key contributions of the paper before responding to reviewers individually.

### Contributions
In this paper, we use a novel Bayesian optimization (BO) to address the problem of query-efficient, perturbation-parsimonious black-box attack on graph classification models, a setup that is important yet under-studied in literature (to our knowledge we are the first in utilising BO in graph adversarial attack. We are grateful that reviewers (8giU, FwkT) have appreciated this novelty). We consider a range of attack settings (edge perturbation/rewiring, node injection) rarely studied in previous literature, and unveil a number of common and interpretable patterns in adversarial examples.

---

### Decision · Program_Chairs · 2021-09-28

**Decision:**

Accept (Poster)

**Comment:**

The reviewers found the exploration of victim models too limited, and experiments should include at least one or two of the state-of-the-art models in the experiments.  Many of the more recent techniques include training with perturbations to the graph structure, and it doesn't seem as if one can say with confidence, without experimental support, that performance improvements with GCN and GIN as victim models will carry over to recent methods.

Please also take into consideration the revised review which cites several relevant papers that address graph classifications

**Consistency Experiment:**

NeurIPS has a long history of experimentation. In 2014, NeurIPS ran an experiment in which 10% of submissions were reviewed by two independent committees to quantify the randomness in the review process. This year, we repeated a variant of this experiment to see how the quality of the review process has changed over time.  This paper was part of the experiment and was therefore assigned to two committees (consisting of reviewers, an Area Chair, and a Senior Area Chair) that reached independent decisions.  If both committees made the same recommendation, this recommendation was followed. If a single committee recommended acceptance, the paper was accepted (with the exception of a few cases in which the other committee identified what we considered a fatal flaw, e.g., an error in a key result).

This copy’s committee reached the following decision: **Reject**

The other committee assigned to the paper recommended **Accept (Poster)**.  You can find the other set of reviews, along with any follow up discussion with the authors here:
https://openreview.net/forum?id=eXxnkL3QfDY